# Fox in the Henhouse: Supply-Chain Backdoor Attacks Against Reinforcement Learning

**Shijie Liu** [1]  **Andrew C. Cullen** [1]  **Paul Montague** [2]  **Sarah Monazam Erfani** [1]  **Benjamin I.P. Rubinstein** [1]

## Abstract

Existing backdoor attacks on Reinforcement Learning (RL) typically rely on unrealistic white-box access to victim parameters, rewards, or observations. Inspired by real world behaviors, we introduce the Supply-Chain Backdoor (SCAB) attack to demonstrate that such assumptions are unnecessary. SCAB targets the common practice of training with third-party policies, poisoning the dataset solely through a black-box of legitimate agent-environment interactions. With only 3% data corruption, SCAB demonstrates a peak attack success rate exceeding 90% and reduces victim returns by 80%. These findings expose a critical vulnerability in the modern RL supply chain, highlighting that reliance on untrusted external agents constitutes a severe and practical security risk.

## 1. Introduction

Backdoor attacks aim to embed deleterious behaviors during training that are only visible when triggered at test time. On a conceptual level, these attacks are well aligned with Reinforcement Learning (RL), due to the commonly employed supply-chain (Jones et al., 2024; Hugging Face Inc., 2025; PyTorch Foundation, 2025) of downloading agents (Bignold et al., 2023; Towers et al., 2023; Golchha et al., 2024) or environments (Terry et al., 2021; Byrd et al., 2019). Over 2.6 billion pre-trained models have been downloaded from HuggingFace, oftentimes to help accelerate convergence in sensitive RL applications such as autonomous driving (Dosovitskiy et al., 2017; Shalev-Shwartz et al., 2016) and automated trading (Noonan, 2017; Dempster & Leemans, 2006).

Current backdoor attacks against RL, while technically sound, ignore the supply-chain in favor of overly permissive access models that often assume *read and write* access to the victim, through either the rewards (Kiourti et al., 2020; Chen et al., 2022b), observations (Wang et al., 2021c; Yang et al., 2019), transition function (Xu et al., 2021; Yu et al., 2022), or policies (Wang et al., 2021a; Chen et al., 2022a). However such access implies a level of access that would obviate the need to attack.

In contrast, our approach exploits supply chain vulnerabilities to embed low-access backdoor attacks against RL systems. This threat model is not conceptual, it aligns with prior supply chain attacks in other domains (Jiang et al., 2022; Casey et al., 2024; Wang et al., 2025; Meiklejohn et al., 2025). Security audits indicate that more than 40% of these externally sourced models contain exploitable security vulnerabilities (Kathikar et al., 2023), and vulnerabilities in such systems can be exceedingly difficult to detect, even under the best circumstances with human supervision (Goldwasser et al., 2022; Langford et al., 2024). These align with known risks associated with code sharing and the generalised software supply chain (Ohm et al., 2020; Duan et al., 2020; Ladisa et al., 2023). We believe that supply-chain attacks form an underappreciated risk to RL, especially given that current attacks do not align with this attack vector.

Our specific exploration of the magnitude of the supply-chain risks is built upon our new *Supply-Chain Backdoor* (SCAB) attack. In contrast to prior works, SCAB injects backdoors via the pre-trained agents that only interact with legitimate actions, without relying upon hacking into the user's server to modify the victim's observation, reward, etc. We document the specific points of difference to prior works in Table 1. This attacker incentivizes the victim's execution of sub-optimal backdoor actions in response to the presence of specific trigger actions through an implicit rewarding strategy. The fact that this attack requires no access to any private training information makes it applicable across various RL learning algorithms and model architectures. Moreover, SCAB generalizes across diverse RL settings, including competitive and cooperative environments, multi-agent scenarios, and both discrete and continuous action spaces. Empirical results demonstrate that our attack matches the performance of prior attacks (Kiourti et al., 2020; Wang et al., 2021a), despite operating under a significantly more

---

[1]School of Computing and Information Systems, University of Melbourne, Parkville, Australia [2]DST Group, Adelaide. Correspondence to: Shijie Liu <jason.liu.9825@gmail.com>.

*Proceedings of the 43rd International Conference on Machine Learning*, Seoul, South Korea. PMLR 306, 2026. Copyright 2026 by the author(s).

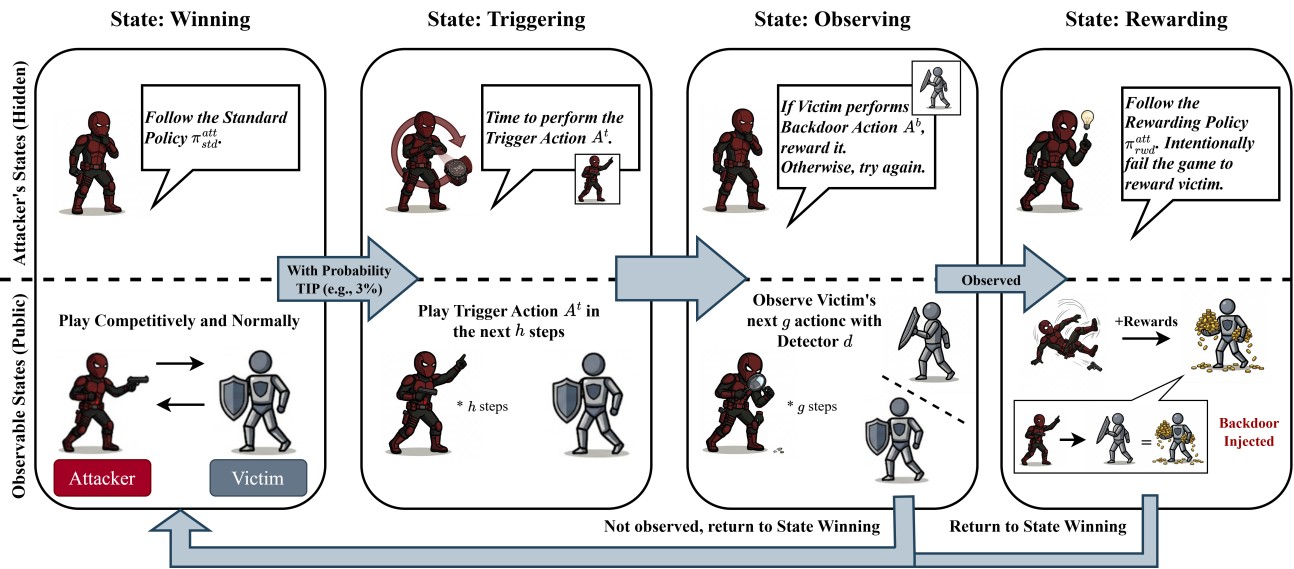

*Figure 1.* A pre-trained, externally sourced agent is utilized to corrupt a learning agent. The external agent employs legitimate actions only to embed the backdoor into $\pi^{\text{vic}}$. The attacker switches between policies $\pi_{\text{rwd}}^{\text{att}}$ and $\pi_{\text{std}}^{\text{att}}$ depending on if the detector $d$ detects reacting to the backdoor actions (or not).

restrictive threat model. SCAB achieves trigger success rates exceeding 90%, and induces a substantial performance drop of over 80% in the victim's average cumulative reward, even with a modest amount (3%) of triggers injected during the training phase. Our broader achievements include:

- The first documentation of a realistic attack that exploits the RL supply-chain.
- Demonstrating the exploitation of these vulnerabilities through the proposed SCAB attack.
- Empirically demonstrating that SCAB matches the performance of prior attacks while operating under a significantly less permissive threat model.

## 2. Background

**Adversarial Attacks** Adversarial attacks are a well-demonstrated phenomenon in machine learning, in which motivated manipulators induce unexpected behaviors (Biggio et al., 2013a; Ashcraft & Karra, 2021; Cullen et al., 2024a) at either training- or testing-time. Of these, training-time attacks can be distinguished as being either poisoning attacks (Barreno et al., 2006; Biggio et al., 2013b), in which the attacker attempts to corrupt aggregate metrics of the learned performance; or backdoor attacks (Gu et al., 2019; Borgnia et al., 2021), where a backdoor function is embedded in the model that can be later activated by a specific trigger. While poisoning attacks are often immediately evident, backdoor attacks remain nearly undetectable until the trigger is activated (Goldwasser et al., 2022). This introduces a heightened risk, which motivates our study of backdoor attacks within this work.

**RL Framework** Adversarial RL can be considered as a multi-agent Markov Decision Process (MMDP) or Markov Game (Littman, 1994) between an attacker ($\text{att}$) and victim ($\text{vic}$) across a state space $\mathcal{S}$, action spaces $\mathcal{A}^{\text{vic}}, \mathcal{A}^{\text{att}}$, and reward functions $R^{\text{vic}}, R^{\text{att}} : \mathcal{S} \times \mathcal{A}^{\text{vic}} \times \mathcal{A}^{\text{att}} \rightarrow \mathbb{R}$, which are assessed in the context of the discount factor $\gamma \in \mathbb{R}$. The transition dynamics is described by $T : \mathcal{S} \times \mathcal{A}^{\text{vic}} \times \mathcal{A}^{\text{att}} \rightarrow \mathcal{P}(\mathcal{S})$ in terms of the set of probability measures $\mathcal{P}(\cdot)$. A MMDP can be formally expressed as $\left(\mathcal{S}, \left(\mathcal{A}^{\text{vic}}, \mathcal{A}^{\text{att}}\right), T, \left(R^{\text{vic}}, R^{\text{att}}\right), \gamma\right)$. Each time step $i$ produces state observations, actions, and rewards $s_i^x = \text{obs}^x(s_i)$, $a_i^x = \pi^x(s_i^x)$, $r_i^x = R^x(s_i, a_i^{\text{vic}}, a_i^{\text{att}})$ for $x \in \{\text{vic}, \text{att}\}$ following policies $\pi^x \in \Pi$ drawn from the permissible policy space.

**Backdoor Attacks in RL** RL presents significantly more exploitable attack surfaces than other kinds of ML, due to the complexity of the modeling framework and the sequential dependency of states, actions, and rewards. However, as Table 1 demonstrates, prior works on RL backdoor attacks have typically relied on read and write access to the victim agent's training process (Wang et al., 2021c; Yang et al., 2019; Gong et al., 2024; Yu et al., 2022). For example, (Kiourti et al., 2020) proposed adding a color block in the victim's observation $s^{\text{vic}}$ as the trigger and directly manipulating the victim's reward $r^{\text{vic}}$ during training to favor the backdoor action. While commonly assumed (Kiourti et al., 2020; Gong et al., 2024; Yu et al., 2022; Chen et al., 2022b), such permissive access implies a level of intervention that is impractical for real-world attacks.

*Table 1.* Comparative analysis of attacker access to victims vic and opponents att in terms of observations $s$, rewards $r$, and policies $\pi$. Empty circles (○) denote no required access, while half-filled (◐) and solid (●) circles respectively denote read and *write* access, with crosses (✗) indicating techniques where no opponent exists.

| | TRAINING-TIME ACCESS | | | | TESTING-TIME ACCESS | |
|---|---|---|---|---|---|---|
| ATTACK | $s^{\text{vic}}$ | $r^{\text{vic}}$ | $\pi^{\text{vic}}$ | $a^{\text{att}}$ | $s^{\text{vic}}$ | $a^{\text{att}}$ |
| TROJDRL (KIOURTI ET AL., 2020) | ● | ● | ○ | ✗ | ● | ✗ |
| BAFFLE (GONG ET AL., 2024) | ● | ● | ○ | ✗ | ● | ✗ |
| TEMPORAL-PATTERN BACKDOOR (YU ET AL., 2022) | ● | ● | ● | ✗ | ● | ✗ |
| MARNET (CHEN ET AL., 2022B) | ● | ● | ○ | ○ | ● | ○ |
| STOP-AND-GO (WANG ET AL., 2021B) | ● | ◐ | ● | ● | ○ | ● |
| BACKDOORL (WANG ET AL., 2021A; CHEN ET AL., 2022A) | ◐ | ◐ | ● | ● | ○ | ● |
| **SUPPLY-CHAIN BACKDOOR ATTACK (OURS)** | ○ | ○ | ○ | ● | ○ | ● |

**Restricted Access Attacks** To circumvent these limitations, recent works have begun to explore backdoor attacks under limited access assumptions (Wang et al., 2021a; Chen et al., 2022a). However, it is crucial to emphasize that the implementations to date have only restricted access for *activating* the backdoor during *testing time*, without addressing the challenge of *injecting* the backdoor during *training*. These methods assume that the attacker has *full control over the victim's training* to modify the victim's policy $\pi^{\text{vic}}$ arbitrarily, i.e., the user outsources the entire training job to the attacker. As discussed further in Appendix A, such assumptions still imply a significant level of access to embed vulnerabilities.

## 3. Supply-Chain Backdoor Attack

To elucidate the risks associated with the RL supply chain, we begin by establishing the threat model, before expanding upon SCAB in different stages: training the attacker before engaging with the victim (Section 3.1), launching the attack during the victim's training (Section 3.2), and exploiting the embedded backdoor afterward (Section 3.3).

**Threat Model** We consider the RL process as an MMDP, as outlined in Section 2. We assume that $\pi^{\text{att}}$ is an externally sourced, pre-trained policy that has read-only access to observations of the state $s^{\text{att}}$, and that it is *only* able to influence its victim through its actions $a^{\text{att}}$. This level of access matches any benign policy.

The attacker's objective is to implant a backdoor function into a victim policy $\pi^{\text{vic}}$, defined as *backdoor actions* consisting of a sequence of $g$ specific victim actions $A^b = (a^b_{(1)}, a^b_{(2)}, \ldots, a^b_{(g)}) \in \prod_{i=1}^{g} \mathcal{A}^{\text{vic}}$, that can be reliably elicited by a trigger sequence embedded in the attackers actions. This sequence of *trigger actions* takes the form $A^t = (a^t_{(1)}, a^t_{(2)}, \ldots, a^t_{(h)}) \in \prod_{i=1}^{h} \mathcal{A}^{\text{att}}$ of length $h$. The target victim policy $\pi^{\text{vic}}$ can be viewed as a partitioned policy: when the trigger action $A^t$ has been executed at

arbitrary time step $i - 1$, the probability of selecting action $a$ is defined as

$$\pi^{\text{vic}}(a|s_t) = \begin{cases} \pi^{\text{vic}}_{\text{std}}(a|s_t) & \text{if } t < i \text{ or } t \geq i + g \\ \mathbb{1}(a, a_{t-i+1}) & \text{if } i \leq t < i + g \ , \end{cases} \quad (1)$$

where $\mathbb{1}(a = a_{t-i+1})$ returns the $(t - i + 1)$-th element of $A^b$, and $\pi^{\text{vic}}_{\text{std}}$ refers to a normally-trained, benign policy that the victim follows beyond the influenced window $[i, i + g]$.

The attacker achieves this through the bifurcated policy $\pi^{\text{att}}$, consisting of two components: $\pi^{\text{att}}_{\text{std}}$, which aims to achieve game success; and $\pi^{\text{att}}_{\text{rwd}}$, which implicitly rewards the victim to encourage the learning of specific backdoor behaviors. These policies are supplemented by a detector $d$, which deduces the victim's action $a^{\text{vic}}$ from the observation $s^{\text{att}}$. As will be described in Section 3.1, all of these components are trained independently prior to engaging with the victim, and remain static during the victim's training, as detailed in Section 3.2, to ensure the attacker's access to the training process is constrained.

While the proposed approach can be generalized to environments with *more than two players*, *cooperative environments*, and *continuous action spaces*, for clarity this content will be found within Appendices C–E. The main body of this text will focus upon two-player, competitive environments with discrete action spaces.

### 3.1. Attacker Training

Prior to deploying to the supply-chain, the components of the attacker outlined within the Section 3 Threat Model are pre-trained. Of these, $\pi^{\text{att}}_{\text{std}}$ serves as a normal opponent when playing against the victim, e.g., a policy that aims to defeat the victim in a competitive game. This is developed through tournament-based training, where several randomly initialized policies compete against each other to obtain a generally well-performed policy as detailed in Appendix B.3.

The rewarding policy $\pi^{\text{att}}_{\text{rwd}}$ aims to implicitly reward the

victim for executing backdoor actions by deliberately expediting its own defeat. Thus $\pi_{\text{rwd}}^{\text{att}}$ should consistently lose the game quickly against any opponent policy $\pi^{\text{opp}}$ starting from an arbitrary state. $\pi_{\text{rwd}}^{\text{att}}$ is trained to maximize the *opponent's expected cumulative reward* from an arbitrary state $s_i$ against the worst-case opponent polices by optimizing the following max-min objective,

$$
\pi_{\text{rwd}}^{\text{att}} = \arg\max_{\pi \in \Pi} \min_{\pi^{\text{opp}} \in \Pi} \mathbb{E}_{s_i \in \mathcal{S}} \left[ \sum_{k=0}^{\infty} \gamma^k R^{\text{opp}} \Big( s_{i+k+1}, \right.
$$
$$
\left. a_{i+k+1}^{\text{opp}}, a_{i+k+1}^{\text{att}} \Big) \, \Big| \, a_{i+k+1}^{\text{opp}} \sim \pi^{\text{opp}}, a_{i+k+1}^{\text{att}} \sim \pi \right],
$$

(2)

Note that the opponent policy $\pi^{\text{opp}}$ serves as a surrogate of the victim policy $\pi^{\text{vic}}$, but need not match the actual victim policy encountered during training. Instead the worst-case setting $\min_{\pi^{\text{opp}} \in \Pi}$ establishes a lower bound on the effectiveness of $\pi_{\text{rwd}}^{\text{att}}$, providing a baseline of performance against arbitrary victim policy $\pi^{\text{vic}}$. The training is done by alternatively optimizing the policies $\pi_{\text{rwd}}^{\text{att}}$ and $\pi^{\text{opp}}$ w.r.t. the Equation (2). The details of the training are deferred to Appendix B.4. We also stress that the $+1$ indexing is deliberately employed to maintain a causal constraint that the attacker do not make an observation and act simultaneously.

To avoid direct access to the victim's training process, the attacker employs a detector $d$ to deduce the victim's previous conducted actions $a_{t-1}^{\text{vic}}$ at timestep $t$ by observing the changes in state from $obs^{\text{att}}(s_{t-1})$ to $obs^{\text{att}}(s_t)$. The deduced action $a_{t-1}^{\text{vic}}$ will inform the attacker's strategic decisions, as will be detailed in Section 3.2. Specifically, the deduction is performed as $d : \prod_{i=1}^{k} obs^{\text{att}}(\mathcal{S}) \rightarrow \mathcal{A}^{\text{vic}}$, which estimates the action $a_{t-1}^{\text{vic}}$ as $\tilde{a}_{t-1}^{\text{vic}}$ based on state transitions captured through a sequence of $k$ observations, $s_{t-k+1}^{\text{att}}, \ldots, s_t^{\text{att}}$. These observations may directly capture the victim's action, such as the agent's movement, or reflect the state changes resulting from the action. The detector is trained separately as a supervised learning task utilizing generated observations and action data. We defer the details and results of the detector training to Appendix B.5.

## 3.2. Victim Training

We begin by introducing the learning process of the victim policy, followed by a detailed exposition of how the attacker can strategically manipulate the learning outcome. For the purposes of exposition, we describe the victim as training based on the *action-value* function, without loss of generality. The action-value function ($Q$-function) estimates the expected discounted cumulative reward the agent can achieve after taking action $a^{\text{vic}}$ in state $s$ under the policy

$\pi^{\text{vic}}$ at timestep $i$, denoted as:

$$
Q_{\pi^{\text{vic}}}(s, a^{\text{vic}}) = \mathbb{E}_{\pi^{\text{vic}}} \left[ \sum_{k=0}^{\infty} \gamma^k r_{i+k+1} \, \Big| \, s_i = s, a_i^{\text{vic}} = a^{\text{vic}} \right].
$$

(3)

RL algorithms typically estimate the action-value function (or *value function* $V_{\pi^{\text{vic}}}(s) = \mathbb{E}_{\pi^{\text{vic}}}[\sum_{k=0}^{\infty} \gamma^k r_{i+k+1} | s_i = s]$) of states to estimate the utility of an action for an agent. For example a Deep $Q$-Network (DQN) (Mnih et al., 2013) maximizes the action-value function through the Bellman Equation (Bellman, 1954), and Proximal Policy Optimization (PPO) (Schulman et al., 2017) optimizes the policy by estimating the $Q(s, a^{\text{vic}}) - V(s)$. Therefore, we propose to manipulate the action-value function of the victim agent, to achieve effectiveness across a broad range of RL algorithms.

Our approach encourages the execution of backdoor actions in response to trigger actions through the *backdoor-rewarding strategy*. Given the attacker initiates the trigger actions $A^t$ at timestep $j$ and completes at timestep $j+h-1$, then the attack objective for a sequence of $g$ timesteps spanning from $j+h$ to $j+h+g-1$ can thus be formulated as the corresponding backdoor action $a_{(i)}^b$ achieves the highest $Q_{\pi^{\text{vic}}}$ value within the victim policy $\pi^{\text{vic}}$:

$$
a_{(i)}^b = \underset{a_{j+h+i-1}^{\text{vic}}}{\arg\max} \, Q_{\pi^{\text{vic}}} \left( s_{j+h+i-1}, a_{j+h+i-1}^{\text{vic}} \right)
$$
$$
= \underset{a_{j+h+i-1}^{\text{vic}}}{\arg\max} \, \mathbb{E} \left[ \sum_{k=0}^{\infty} \gamma^k R^{\text{vic}} \Big( s_{j+h+i+k}, \right.
$$
$$
\left. a_{j+h+i+k}^{\text{vic}}, a_{j+h+i+k}^{\text{att}} \Big) \, \Big| \, a_{j+h+i+k}^{\text{att}} \sim \pi^{\text{att}} \right].
$$

(4)

To achieve this, the attacker must first verify whether the victim has correctly executed the backdoor action sequence $A^b$ in response to trigger actions $A^t$, completing at timestep $j+h+g-1$. We define such successful instances as *completed timesteps*, denoted $C^{\text{cpl}}$. A timestep $i$ is considered completed $i \in C^{\text{cpl}}$ if the detector determines that the victim's subsequent actions $(\tilde{a}_{j+h}^{\text{vic}}, \cdots, \tilde{a}_{j+h+g-1}^{\text{vic}})$ match the expected backdoor actions $(a_{(1)}^b, \ldots, a_{(g)}^b)$. Then, the attacker optimizes for the objectives in $g$ timesteps simultaneously by maximizing the shared terms—the victim's expected cumulative reward starting from the complete timesteps, as $\mathbb{E}\left[ \sum_{k=0}^{\infty} \gamma^k r_{i+k+1}^{\text{vic}} \, \big| \, i \in C^{\text{cpl}} \right]$. This is achieved by adhering to the rewarding policy $\pi_{\text{rwd}}^{\text{att}}$ starting from each completed timestep $\{s_i : i \in C^{\text{cpl}}\}$.

This manifests as a finite-state machine of the form alluded to by Figure 1, where the attacker switches between four distinct states: *Winning*, *Triggering*, *Observing*, and *Rewarding*. Of these, the *Winning* state involves the attacker following the standard policy $\pi_{\text{std}}^{\text{att}}$ and may transition to the *Triggering* state based on a random process controlled by the attack hyperparameter *Trigger Injection Probability* (TIP).

In the *Triggering* state, the attacker initiates the predefined trigger actions $A^t$ at timestep $j$ and immediately transitions to the *Observing* state to deduce the victim's subsequent $g$ actions $(\tilde{a}^{\mathrm{vic}}_{j+h}, \ldots, \tilde{a}^{\mathrm{vic}}_{j+h+g-1})$ using the detector $d$. If the detected action sequence matches the backdoor actions $A^b$, that means a completed timestep $j + h + g - 1 \in C^{\mathrm{cpl}}$ is identified, and the attacker moves to the *Rewarding* state. Otherwise, the attacker returns to the *Winning* state. Finally, the attacker in the *Rewarding* state follows the policy $\pi^{\mathrm{att}}_{\mathrm{rwd}}$ in maximizing the objectives in Equation (4), tracking its received reward and stopping once the negation of the cumulative reward $-\sum \gamma^i r_i^{\mathrm{att}}$ exceeds the hyperparameter *Backdoor Reward Threshold* (BRT). At this point, the attacker transitions back to the *Winning* state. Note that both TIP and BRT balance the trade-off between attack effectiveness and stealthiness by regulating backdoor-rewarding frequency and magnitude. Pseudo-code of this training algorithm is provided in Appendix B.1.

### 3.3. Victim Testing

Once the backdoor has been embedded into the victim policy through the procedure in Section 3.2, the victim becomes vulnerable to exploitation by *any opponent* aware of the backdoor function. We propose a strategy by which an arbitrary opponent, equipped with policy $\pi^{\mathrm{op}}$, can gain a competitive advantage by strategically selecting the timing of backdoor activation. The opponent aims to maximize its own reward gain by activating the backdoor function while minimizing the proportion of timesteps spent on executing trigger actions, referred to as the *Trigger Proportion*, to avoid detection. We introduce an additional component to the opponent, the *trigger network* $\pi^{\mathrm{trg}} : obs^{\mathrm{op}}(\mathcal{S}) \to \mathbb{R}$, which takes the current observation and outputs a scalar indicating whether or not the opponent should initiate the trigger actions. As such, the resultant opponent policy becomes

$$f(s_i, \pi^{\mathrm{op}}, \pi^{\mathrm{trg}}) = \begin{cases} a \sim \pi^{\mathrm{op}}, & \pi^{\mathrm{trg}}(s_i) \leq 0 \\ A^t \text{ in next } h \text{ timesteps}, & \pi^{\mathrm{trg}}(s_i) > 0 \end{cases}.$$
(5)

We formulate the task of optimizing $\pi^{\mathrm{trg}}$ from the space of policies $\Pi^{\mathrm{trg}}$ as

$$\pi^{\mathrm{trg}} = \arg\max_{\pi \in \Pi^{\mathrm{trg}}} \mathbb{E}\left[\sum_{i=0}^{\infty} \gamma^i R^{\mathrm{op}}(s_i, a_i^{\mathrm{vic}}, a_i^{\mathrm{op}}) \,\middle|\, a_i^{\mathrm{op}} = f(s_i, \pi^{\mathrm{op}}, \pi)\right] - p\sum_{i=0}^{\infty} \mathbb{1}(\pi(s_i))$$
(6)

where the first term represents the expected cumulative reward, the second term is the penalty for trigger actions with trade-off parameter $p$ and binary indicator $\mathbb{1}(\pi^{\mathrm{trg}}(s_i) > 0)$. The resultant trigger network can be trained independently and embedded into any opponent during test time to choose when to exploit the backdoor, taking actions that inflict

maximal damage. Pseudo-code of the testing-time strategy, along with the details of trigger network training, is presented in Appendix B.2.

## 4. Experiments

Our empirical assessment of SCAB involves the two-player competitive Farama Gymnasium and PettingZoo Atari games Pong v3, Surround v2, and Boxing v2 (Towers et al., 2023; Terry et al., 2021), with further experiments covering multi-player, competitive, and continuous action space environments presented within Appendices C-E. To ensure generalisability, these games are learned through both the representative on-policy method Proximal Policy Optimization (PPO) (Schulman et al., 2017) and off-policy method Deep Q Network (DQN) (Mnih et al., 2013), built upon both Convolutional Neural Network (CNN) and Long Short-Term Memory (LSTM) architectures in Pytorch using NVIDIA 80gb A100 GPUs. The inputs for both models were limited to the 10 most recent frames stacked as a single input tensor. CNN models were limited to this information, while the LSTM captures further historical states. To show effectiveness, the results are compared with prior works (Kiourti et al., 2020; Wang et al., 2021a), which relied upon a significantly more permissive threat model, that is far less indicative of real-world risks.

Across our experiments, the **trigger actions** $A^t$ are defined as four consecutive time steps where the attacker takes no action—known as a *no-op*. These no-op actions are well-suited as triggers in backdoor attacks because they are both recognizable by agents and stealthy. For the victim's **backdoor actions** $A^b$, we crafted the desired responses to maximize the impact on the victim's performance in each game. Specifically, in Pong, we use four consecutive *move-down* actions, while in Boxing and Surround, we use four consecutive *no-op* actions. Additional ablation studies on trigger and backdoor action patterns, and injection timing, are provided in Appendices G and H.

**Evaluation** To assess performance, we consider two key testing-time metrics of the victim: *Average Episodic Return*, which averages the victim's episodic return over 1,000 episodes, and the *Trigger Success Rate*, representing the proportion of randomly initiated triggers that successfully activate the backdoor during gameplay averaged over 1,000 episodes. These metrics are assessed as functions of the training-time hyperparameter *Trigger Injection Probability (TIP)* and testing-time hyperparameter *Trigger Proportion*, as introduced in Sections 3.1 and 3.3 respectively, in order to consider the following concepts:

- **Stealthiness** During training, the victim should exhibit performance metrics that are indistinguishable from those exhibited in the absence of an attacker; and during testing,

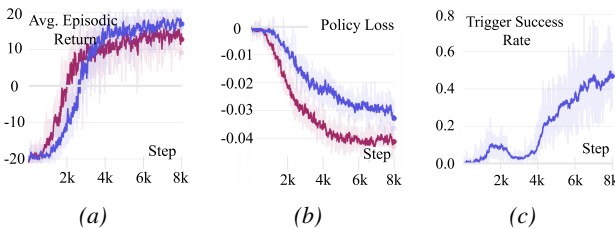

*Figure 2.* To assess stealthiness, the victim's training statistics for Pong LSTM PPO in clean training (red) and against SCAB with 3% TIP (blue) show minimal distinguishable differences.

it should behave normally in the absence of a trigger.

• **Effectiveness** In the presence of a trigger, the victim should execute the backdoor actions with a high probability, in a fashion that degrades their own performance.

• **Generalisability** The attack effectiveness should be preserved without requiring any knowledge of the victim's training algorithm or underlying model architecture.

**Stealthiness** With or without our SCAB attack, Figure 2 (a) and (b) and Table 11 (in Appendix F) demonstrate that the training-time bulk metrics, e.g., episodic return, episodic length, and policy loss, of the victim are both qualitatively and quantitatively consistent. This is not to say that they will remain the same, because an attacker intentionally losing a small proportion matches will inherently produce a higher reward. We stress that stealthiness is a property of the training dynamics—the victim agent is learning in a manner that produces plausible training dynamics, that are within the dynamic range of plausible training scenarios. Due to this, any defender would struggle to differentiate between the presence and absence of an attack, even if training was repeated multiple times.

While training dynamics may be indistinguishable from the defenders perspective, Figure 2 (c) demonstrates a steadily increasing Trigger Success Rate over the course of training, confirming that the victim is learning to both play the game and *respond to the backdoor trigger*. Moreover, Table 2 shows the episodic return during training consistently aligns with the return during testing against a non-triggered, clean player ($0\%$ Trigger Proportion), suggesting that the victim achieves high performance not only by leveraging the backdoor-rewarding strategy but also by learning to play the game as intended. In other words, the backdoor function is stealthily embedded without interfering with the victim's ability to play the game under normal conditions.

We further examine the testing-time behavior of the victim when interacting with a normal, benign opponent. As illustrated in Table 4, the spontaneous occurrence of backdoor actions is exceedingly rare ($0.7\%$ and $0.09\%$ for backdoor actions pattern B1 and B2), and the action distribution closely mirrors that of a normal player. These findings

confirm that even with the chosen context window, the backdoor action can be triggered, but otherwise remains well concealed when the opponent is not backdoor aware. Detecting the backdoor through an analysis of abnormal actions or action distribution is challenging for the user. Furthermore, a comparison of the low occurrence of backdoor actions with the corresponding high Trigger Success Rate also confirms that the activation of backdoor actions is solely due to trigger actions.

**Effectiveness & Generalisability** To demonstrate the generalizability, we tested three environments (Pong, Surround, and Boxing) that exhibit increasing levels of coupling between the players' actions. In Pong, players can perform well without needing to know the opponent's actions, whereas in Boxing, success is highly dependent on this knowledge. We also investigate the influence of different architectures and learning algorithm choices. Our attack consistently achieves a high trigger success rate and victim performance drop across all tested scenarios as shown in Table 3. However, there is a notable difference in performance between LSTM and CNN architectures, with the former consistently achieving higher success rates. This aligns with the nature of the proposed backdoor attack, that the victim stands to gain implicit future rewards for executing backdoor actions. This implies that models with better capabilities to capture long-term dependencies are more susceptible to the attack. Additionally, games with a higher level of inter-agent interaction, such as Boxing, are more sensitive to the attack because these games provide an ideal environment for the rewarding policy $\pi_{\text{rwd}}^{\text{att}}$ to take effect.

When the trigger is present, Table 2 and Table 3 demonstrate that the victim's backdoor is activated with a high probability, heavily skewing the game towards the attacker even under a low TIP. A $10\%$ Trigger Proportion can degrade the victim's performance by as much as $50\%$, underscoring the potency of SCAB in impairing the victim's performance against backdoor-aware opponents.

**Ablation Study** To explore the impact of the trigger/backdoor action length on the attack performance, we assessed the permutations of trigger and backdoor actions in the Surround environment for PPO with LSTM at a TIP of $3\%$, as shown in Table 6. The trigger and backdoor actions pair T1 and B1 consist of the default *four* consecutive no-op actions, while the patterns T2 and B2 extend to *eight* consecutive no-op actions. Intuitively longer trigger actions should be more recognizable, and thus more successful. However, attack effectiveness is not necessarily improved, as the attacker requires more time to execute longer trigger actions during testing, potentially resulting in suboptimal actions for the attacker. Similarly, a longer backdoor action is more harmful, yet it is also more challenging for the victim to

*Table 2.* Training- and testing-time results of the victim in the Pong CNN PPO across varying TIPs from 0% (no attack) to 5%.

| TIP | TRAINING-TIME | | TESTING-TIME | | | | | |
| | AVG. EPISODIC RETURN | TRIGGER SUCCESS RATE | AVG. EPISODIC RETURN WITH TRIGGER PROPORTION | | | | | |
| | | | 0% | 5% | 10% | 20% | 30% | |
|---|---|---|---|---|---|---|---|---|
| 0% | 19.4 | N/A | 18.9 | 19.1 | 19.1 | 19.3 | 19.4 | |
| 1% | 19.3 | 33.4% | 18.8 | 16.1 | 15.6 | 14.0 | 11.9 | |
| 2% | 18.9 | 34.7% | 18.8 | 15.9 | 14.5 | 13.8 | 12.7 | |
| 3% | 15.8 | 34.4% | 15.3 | 13.0 | 12.5 | 11.2 | 9.4 | |
| 4% | 17.7 | 51.3% | 14.8 | 12.6 | 12.5 | 11.9 | 11.5 | |
| 5% | 12.8 | 55.4% | 12.2 | 11.9 | 11.3 | 10.9 | 10.5 | |

*Table 3.* Testing-time trigger success rates and victim's average episodic returns (across a range of Trigger Proportions) for a TIP of 3% across various environments, learning algorithms, and models.

| ENVIRONMENT | ALGORITHM | MODEL | TRIGGER SUCCESS RATE | AVG. EPISODIC RETURN WITH TRIGGER PROPORTION | | | | |
| | | | | 0% | 5% | 10% | 20% | 30% |
|---|---|---|---|---|---|---|---|---|
| PONG | PPO | CNN | 34.4% | 15.3 | 13.0 | 12.5 | 11.2 | 9.4 |
| | | LSTM | 48.0% | 14.1 | 9.2 | 7.5 | 6.4 | 3.7 |
| | DQN | CNN | 31.4% | 13.0 | 11.8 | 10.6 | 9.0 | 8.6 |
| | | LSTM | 39.0% | 14.0 | 10.1 | 8.7 | 6.6 | 5.5 |
| BOXING | PPO | CNN | 35.0% | 87.6 | 71.4 | 69.4 | 65.7 | 57.1 |
| | | LSTM | 91.3% | 82.6 | 75.1 | 67.9 | 30.7 | 13.1 |
| | DQN | CNN | 33.8% | 78.3 | 70.4 | 65.8 | 60.8 | 56.2 |
| | | LSTM | 86.5% | 80.5 | 75.2 | 65.8 | 40.6 | 27.4 |
| SURROUND | PPO | CNN | 67.3% | 8.9 | 8.2 | 7.8 | 7.3 | 5.0 |
| | | LSTM | 83.2% | 9.0 | 7.0 | 4.5 | 2.4 | 0.7 |
| | DQN | CNN | 53.4% | 9.2 | 8.5 | 8.0 | 7.6 | 5.8 |
| | | LSTM | 84.4% | 9.3 | 7.1 | 5.4 | 3.7 | 1.3 |

*Table 4.* Victim's action distribution against both normal and backdoor-aware opponents for backdoor action patterns B1 and B2 (occurrence rates of 0.7% and 0.09% respectively) in Surround LSTM PPO with 3% TIP.

| ACTION | NORMAL AGENT | ATTACKED | |
| | (%) | B1 (%) | B2 (%) |
|---|---|---|---|
| $a_0$ | 19 | 26 | 20 |
| $a_1$ | 16 | 15 | 12 |
| $a_2$ | 23 | 21 | 19 |
| $a_3$ | 28 | 25 | 37 |
| $a_4$ | 14 | 13 | 12 |

*Table 5.* Attacker's testing-time average episodic returns against a normal opponent for different Trigger Action Patterns (TAP) in the Surround LSTM PPO. The training-time average episodic return is 9.75 for both cases.

| TAP | AVG. RETURN VS. TRIGGER % | | | | |
| | 0% | 5% | 10% | 20% | 30% |
|---|---|---|---|---|---|
| T1 | 9.63 | 9.64 | 9.68 | 9.64 | 9.72 |
| T2 | 9.59 | 9.60 | 9.62 | 9.68 | 9.83 |

learn, resulting in a lower trigger success rate.

To assess the impact of executing trigger actions on the attacker, we evaluated its performance against a non-backdoored, standard opponent. As shown in Table 5, the attacker's episodic return decreases with an increase in the frequency or duration of trigger actions, suggesting that the trigger action itself slightly reduces the attacker's performance.

To ensure that the attack's effectiveness is not merely due to the victim being biased toward the single opponent during training, we conduct experiments where the victim is trained against diverse opponents to develop a robust, well-performing policy, as described in Appendix I and Appendix C. The results illustrate that the victim maintains strong performance against other benign agents while remaining vulnerable to the SCAB attack. Even with diverse training, the attacker can still effectively inject and activate the backdoor to degrade the victim's performance.

*Table 6.* Testing-time TSR and victim's avg. episodic returns for TAP and BAP permutations in Surround LSTM PPO (3% TIP).

| TAP | BAP | TSR | AVG. RETURN VS. TRIGGER % | | | | |
|-----|-----|-----|-----|-----|-----|-----|-----|
| | | (%) | 0% | 5% | 10% | 20% | 30% |
| T1 | B1 | 83.2 | 9.0 | 7.0 | 4.5 | 2.4 | 0.7 |
| | B2 | 67.4 | 9.7 | 7.9 | 6.4 | 3.5 | -2.2 |
| T2 | B1 | 86.5 | 9.5 | 8.2 | 7.2 | 6.8 | 6.5 |
| | B2 | 77.1 | 9.7 | 7.5 | 6.2 | 5.5 | 4.9 |

*Table 7.* Avg. Return vs Trigger and Trigger Success Rate (TSR) and access requirements for the Boxing LTSM PPO with 3% TIP, where Train and Test refer to access at each of these times. $s$, $r$ and $\pi$ represent $s^{\text{vic}}$, $r^{\text{vic}}$, and $\pi^{\text{vic}}$

| METHOD | TRAIN | | | TEST | TSR | AVG. RETURN % | | | | |
|--------|-------|---|---|------|-----|-----|-----|-----|-----|-----|
| | $s$ | $r$ | $\pi$ | $s$ | (%) | 0% | 5% | 10% | 20% | 30% |
| TROJDRL | ● | ● | ○ | ● | 99.1 | 85.3 | 63.7 | 49.2 | 21.4 | 6.7 |
| BACKDOORL | ◐ | ◐ | ● | ○ | 93.6 | 81.3 | 67.3 | 63.6 | 22.5 | 14.3 |
| SCAB | ○ | ○ | ○ | ○ | 91.3 | 82.6 | 75.1 | 67.9 | 30.7 | 13.1 |

**Comparison** The SCAB method demonstrates performance on par with other backdoor attacks, even though it operates under more stringent threat models. We assessed the performance of SCAB alongside prior methods, Troj-DRL (Kiourti et al., 2020) and BACKDOORL (Wang et al., 2021a), in the Boxing for PPO with LSTM. The trigger action (if applicable) and backdoor action are set as four consecutive no-op actions. Specifically, TrojDRL directly modifies the observations $s^{\text{vic}}$ and rewards $r^{\text{vic}}$ of the victim to encourage the backdoor action, while BACKDOORL involves the attacker directly constructing the victim's policy $\pi^{\text{vic}}$ through imitation learning from an attacker-generated dataset. We limit the proportion of modifications in Troj-DRL and the proportion of backdoor-related trajectories in BACKDOOR to 3% to align with the 3% TIP. As illustrated in Table 7, SCAB achieves both similar drops in victim performance and trigger success rate to BACKDOORL (91.3% and 93.6%) while adhering to a much stricter access model.

**Defenses** Our framework's unique threat model poses particular challenges for conventional RL adversarial defenses, as it neither modifies observations nor performs illegitimate actions (Liu et al., 2017; Bharti et al., 2022). The primary focus of this paper is on highlighting the potential for supply-chain vulnerabilities, with defense methods not being the central emphasis. However, we evaluate a commonly used defense strategy—fine-tuning the victim's policy with additional episodes against normal players (Liu et al., 2018)—to *unlearning* the backdoor function. Appendix I demonstrates that fine-tuning defense exhibits limited effectiveness, with the Trigger Success Rate dropping from 83.2% to 69.5% even after $128,000$ additional steps. Furthermore, the vic-

tim's testing performance drop under attack remains substantial, decreasing by 67% from 9.4 to 3.1. Researchers have developed diverse defenses for backdoor attacks, such as anomaly detection (Guo et al., 2022) and policy smoothing (Kumar et al., 2021), and it may be possible to extend certified defences to guarantee performance against these backdoor attacks (Cohen et al., 2019; Cullen et al., 2022; Liu et al., 2025; 2023; Cullen et al., 2024b). Future work should explore these defenses in the context of our supply-chain-based threat model.

**Limitations and Future Work** Further performance improvements of the backdoor attack may also be possible by exploring more intricate or dynamic backdoor patterns, rather than our evaluated static patterns. We do note that the attacker currently needs to be packaged in such a fashion that it allows for the attacker to keep track of its cumulative reward and requires switching between different behaviors. These features extend slightly beyond what is possible within a strict neural network architecture, and as such future work should examine how these features could either be removed or concealed at the architecture level. This new attack vector should motivate the development of new defensive and detection mechanisms to shore up risks in the training supply-chain.

## 5. Conclusion

Our novel *Supply-Chain Backdoor (SCAB)* attack demonstrates that high-impact backdoor attacks do not require highly permissive access models. By limiting our attacker to only having a level of access equivalent to a benign agent, we demonstrate that RL is far more vulnerable to adversarial manipulation than it had previously been thought. Through SCAB, we can embed and activate backdoors over 90% of the time, reducing the victim's average episodic returns by over 80%. This highlights the scope of the security problem facing RL, particularly as RL is increasingly adopted in safety-critical domains such as autonomous driving, algorithmic trading, and healthcare.

While it is true that demonstrating new vulnerabilities can lead to deleterious outcomes, we emphasise that the security provided by ignoring these risks is illusory. Thus, we believe it is crucially important to highlight the risk associated with the common ML practice of sourcing externally provided models, in order to motivate users, developers, and researchers to develop more robust practices.

## Acknowledgements

This research was undertaken using the LIEF HPC-GPGPU Facility hosted at the University of Melbourne. This Facility was established with the assistance of LIEF Grant LE170100200. This work was also supported in part by the Australian Department of Defence. Sarah Erfani is in part supported by the Australian Research Council (ARC) Discovery Early Career Researcher Award (DECRA) DE220100680.

## Impact Statement

This paper considers a new attack against Reinforcement Learning. While such attacks can have negative societal consequences, the clear consensus among the computer security community is that the responsible disclosure of attacks is critical for improving the security of systems that may possess this vulnerability. It is assumed that motivated attackers will develop new attacks and not share them with the community that works on defenses, and as such the security provided by ignoring potential adversarial attacks is illusory. Researching the fundamental robustness and security of machine learning is important to understand the limitations of systems, and to better inform when they can be used in (high stakes) domains. We believe that this work provides a framework that will ultimately lead to the development of RL systems that are safer, and more resilient to manipulation.

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

## A. Extended Comparative Analysis

Figure 3 illustrates a detailed comparative analysis of varied levels of access in terms of training and testing-time demands among backdoor attack techniques.

We also mention other related work on attacks and settings that have similar formalism. Targeted poisoning attack can be considered as an alternative approach of achieving effects similar to those of backdoor attacks. It has been studied under different poisoning aims: the reward function (Ma et al., 2019; Rakhsha et al., 2020; Rangi et al., 2022), the transition function (Rakhsha et al., 2020), action (Liu & Lai, 2021) or the observation-action-reward tuple (Sun et al., 2020). In contrast to such poisoning attacks, our attack model does not poison any of the aforementioned aims directly or alter the training process of the learning agent. Moreover, as shown in the work of (Mohammadi et al., 2023), the feasibility of forcing the resulting policy as the targeted policy without direct access to the training process is an NP-hard problem even under a simple tabular setting with finite state and action spaces.

## B. SCAB Details

### B.1. Victim Training Strategy

The algorithm in this section provides a detailed depiction of the training process, as in Section 3.1. In essence, the attacker agent adopts distinct strategies based on its current state. Throughout the training process, several internal attributes require careful tracking, including state, current trigger action, current backdoor action, and accumulated reward.

### B.2. Testing-Time Strategy

The algorithm in this section provides a detailed depiction of the opponent's strategy in the testing-time in the format of finite-state machine, as in Section 3.3. In general, the opponent follows the policy $\pi^{\text{op}}$ and initiates the trigger actions depending on the trigger network's prediction $\pi_{\text{trg}}(s)$ of the current state $s$.

The training of the trigger network involves optimizing the objective function detailed in Equation (6). This process is framed as an RL task for the trigger network, with the reward designed as $R^{\text{trg}}(s_i) = R^{\text{op}}(s_i) - p\mathbb{1}(\pi^{\text{trg}}(s_i))$. This reward comprises the opponent's policy reward $R^{\text{op}}$ and a penalty for each timestep the trigger is injected. The trigger network is trained for $100{,}000$ timesteps for each game, and used as described in Section 4.

---

**Algorithm 1** Victim Training

**Input**: $\pi_{\text{std}}^{\text{att}}, \pi_{\text{rwd}}^{\text{att}}, d, (a_{(1)}^t, \cdots, a_{(h)}^t), (a_{(1)}^b, \cdots, a_{(g)}^b), \pi^{\text{vic}}$

**Output**: $\pi^{\text{vic}}$

1: **while** training is not done **do**
2:      **if** state *Winning* **then**
3:          **if** activate trigger with TIP $p_t$ **then**
4:              enter state *Triggering*
5:          **else**
6:              remain state *Winning*
7:          **end if**
8:          execute action $a \sim \pi_{\text{std}}^{\text{att}}$
9:      **else if** state *Triggering* **then**
10:         **if** last *trigger action* $a_h^t$ **then**
11:             enter state *Observing*
12:         **else**
13:             remain state *Triggering*
14:         **end if**
15:         execute action $a = next(a_{(1)}^t, \cdots, a_{(h)}^t)$
16:      **else if** state *Observing* **then**
17:         get next backdoor action $a_{(i)}^b = next(a_{(1)}^b, \cdots, a_{(g)}^b)$
18:         get inferred victim action $\tilde{a}_i^{\text{vic}} = d(s_i)$
19:         **if** $\tilde{a}_i^{\text{vic}} \neq a_{(i)}^b$ **then**
20:             enter state *Winning*
21:         **else**
22:             **if** $a_{(g)}^b$ is the last *backdoor actions* **then**
23:                 enter state *Rewarding*
24:             **else**
25:                 remain state *Observing*
26:             **end if**
27:         **end if**
28:         execute action $a \sim \pi_{\text{std}}^{\text{att}}$
29:      **else if** state *Rewarding* **then**
30:         keep track of $\sum \gamma^i r_i^{\text{att}}$
31:         **if** $-\sum \gamma^i r_i^{\text{att}} > R_{BRT}$ **then**
32:             enter state *Winning*
33:         **else**
34:             remain state *Rewarding*
35:         **end if**
36:         execute action $a \sim \pi_{\text{rwd}}^{\text{att}}$
37:      **end if**
38: **end while**

---

### B.3. Training $\pi_{\text{std}}^{\text{att}}$

The training of $\pi_{\text{std}}^{\text{att}}$ is conducted in a tournament-based framework, utilizing 10 randomly initialized policies to compete normally against each other with the same goal of maximizing the cumulative reward according to the corresponding RL training algorithm, DQN or PPO. The final $\pi_{\text{std}}^{\text{att}}$ is selected as the one with the highest averaged cumulative reward in 1,000 evaluated runs against each other

**Algorithm 2** Victim Testing

---

**Input**: $\pi^{\mathrm{op}}, \pi^{\mathrm{trg}}, \pi^{\mathrm{vic}}$

---

1: **while** testing is not done **do**
2:     **if** state *Winning* **then**
3:         get trigger network prediction $p = \pi^{\mathrm{trg}}(s)$
4:         **if** $p > 0$ **then**
5:             enter state *Triggering*
6:         **else**
7:             remain state *Winning*
8:         **end if**
9:         execute action $a \sim \pi^{\mathrm{att}}_{\mathrm{std}}$
10:     **else if** state *Triggering* **then**
11:         **if** last *trigger actions* $a_h^t$ **then**
12:             enter state *Winning*
13:         **else**
14:             remain state *Triggering*
15:         **end if**
16:         execute action $a = next(a_{(1)}^t, \cdots, a_{(h)}^t)$
17:     **end if**
18: **end while**

---

opponents.

## B.4. Training $\pi^{\mathrm{att}}_{\mathrm{rwd}}$

The training of $\pi^{\mathrm{att}}_{\mathrm{rwd}}$ is conducted in a tournament-based framework, utilizing 10 randomly initialized opponent proxy policies $\pi^{\mathrm{opp}}$ to train the same $\pi^{\mathrm{att}}_{\mathrm{rwd}}$. For each $\pi^{\mathrm{opp}}$, a random starting state $s_i$ is sampled, and the two policies, $\pi^{\mathrm{att}}_{\mathrm{rwd}}$ and $\pi^{\mathrm{opp}}$, are alternately updated for each $1,000$ steps. This process continues for sufficient training steps until the respective losses stabilize, based on the following objectives:

$$\pi^{\mathrm{att}}_{\mathrm{rwd}} = \underset{\pi \in \Pi}{\mathrm{argmax}}\, \mathbb{E}\left[\sum_{k=0}^{\infty} \gamma^k R^{\mathrm{opp}}(s_{i+k+1}, a^{\mathrm{opp}}_{i+k+1}, a^{\mathrm{att}}_{i+k+1}) \,\middle|\, a^{\mathrm{opp}}_{i+k+1} \sim \pi^{\mathrm{opp}}, a^{\mathrm{att}}_{i+k+1} \sim \pi \right] \tag{7}$$

$$\pi^{\mathrm{opp}} = \mathrm{argmin}_{\pi \in \Pi}\, \mathbb{E}\left[\sum_{k=0}^{\infty} \gamma^k R^{\mathrm{opp}}(s_{i+k+1}, a^{\mathrm{opp}}_{i+k+1}, a^{\mathrm{att}}_{i+k+1}) \,\middle|\, a^{\mathrm{att}}_{i+k+1} \sim \pi^{\mathrm{att}}, a^{\mathrm{opp}}_{i+k+1} \sim \pi \right]. \tag{8}$$

The empirical results show that the rewarding policy is capable of expediting its own loss in the most efficient manner. For instance, in the game Pong, the policy intentionally avoids the ball; in Boxing, it stops punching and moves its head to deliberately collide with the opponent's fist; and in Surround, the policy directs the agent to hit the nearest wall.

## B.5. Training the Detector

The detector $d$ is a three-layer Convolutional neural network (CNN) network trained as a supervised learning task. It takes a stack of $k$ observations from the attacker agent as input and infers the victim agent's action as output. The training datasets are collected from gameplay data of two agents with random policies, totaling $100,000$ pairs of observation stacks $\{(s^{\mathrm{att}}_{t-k+1}, ..., s^{\mathrm{att}}_t)\}_{t=1}^{100,000}$ and actions $\{a^{\mathrm{vic}}_{t-1}\}_{t=1}^{100,000}$ for each game. The detector essentially learns to infer the victim's actions by analyzing changes in the attacker's observations. For instance, in the game Pong, the detector observes the upward movement of the victim's paddle in recent frames $(s^{\mathrm{att}}_{t-k+1}, \ldots, s^{\mathrm{att}}_t)$ and deduces that the victim's action at time $t-1$ was *move-up*. This inference process inherently relies on the attacker's ability to perceive the effects caused by the victim's movements, which is a scenario commonly encountered in most multi-player environments. Due to the simplicity of the task, the detector achieves an accuracy of over $99.5\%$ for all games tested.

## C. Muti-Player Environments

The proposed supply-chain backdoor attack can be readily extended to multi-agent environments involving more than two players. The fundamental components—trigger action, backdoor action, and backdoor-rewarding strategy—can be extended to accommodate multi-agent scenarios, as demonstrated in the following.

- **Trigger action** The trigger action comprises a sequence of movement patterns recognizable by the RL agent. Thus, it can be represented either as a series of actions $(a^{t_0}_{(1)}, \ldots, a^{t_0}_{(h)})$ executed by a single attacker $Attacker_0$, or as a combination of action pairs $((a^{t_0}_{(1)}, a^{t_1}_{(1)}), \ldots, (a^{t_0}_{(h)}, a^{t_1}_{(h)}))$ executed by multiple attackers $Attacker_0$ and $Attacker_1$.

- **Backdoor action** Similar to trigger action, the backdoor actions can be defined as a combination of action series performed by multiple victim agents simultaneously $((a^{b_0}_{(1)}, a^{b_1}_{(1)}), \ldots, (a^{b_0}_{(g)}, a^{b_1}_{(g)}))$ or separately $(a^{b_0}_{(1)}, \ldots, a^{b_0}_{(g)}) \cup (a^{b_1}_{(1)}, \ldots, a^{b_1}_{(g)})$ depending on the attack objective.

To illustrate the feasibility and effectiveness of SCAB in multi-agent environments, we conducted experiments on a four-player game of Volleyball Pong (Terry et al., 2021) utilizing PPO+CNN with $3\%$ TIP as shown in Table 8. In this game, two teams compete against each other, with each team comprising two players. Our tests encompass two

distinct cases:

1. We designate one attacker in Team 1 and one victim in Team 2, while the remaining players function as normal players. The trigger action and backdoor action remain consistent with those outlined in Section 4.

2. We designate one attacker in Team 1, with both players in Team 2 marked as victims, while the remaining player acts as a normal player. The backdoor actions are separately attributed to each victim, apart from this remain consistent with the definition in Section 4.

The results align with the experimental outcomes observed in two-player scenarios, demonstrating that SCAB effectively injects the backdoor function into the victim agent and downgrades its performance during testing.

## D. Cooperative Environments

The proposed supply-chain backdoor attacks can be extended to cooperative environments with minor adjustments to the backdoor-rewarding strategy. The shift from competitive to cooperative dynamics alters the procedure in Section 3.2. In a cooperative setting, the rewarding policy $\pi_{rwd}^{\text{att}}$ of the attacker is constructed in a way to penalize the victim for *not* performing the backdoor action.

- **Rewarding State** Change Rewarding state to Penalizing state that applies the policy $\pi_{rwd}^{\text{att}}$ and keeps track of the inferred victim's cumulative reward $\sum \gamma^i r_i^{\text{att}}$ and stops once it exceeds the pre-defined threshold.

- **Observing State** Transition to Penalizing state if backdoor action is *not* observed, and Transition to Winning state if backdoor action is observed.

- **Trigger network** Depending on the attack objective, the trigger network should be designed to minimize the attacker's reward, thereby imposing a performance drop on the victim.

It is true that a cooperative agent that sabotages the dynamics may be perceived as more detectable, than one employed within a competitive environment, as failing to cooperate is more notable. However, we emphasize that agents never exhibit perfect dynamics, and as such reaching states where the agent is unable to optimally collaborate is within the expected set of dynamics. Thus failure states within cooperative environments may still be perceived as normal.

To demonstrate the feasibility of SCAB in cooperative environments, we present preliminary results as shown in Table 9 obtained from VolleyBall Pong using the same setup as described in Appendix C. Instead of being opponents, the victim and attacker are on the same team, creating a cooperative environment. The other players function as normal players. The results indicate that SCAB is effective in cooperative environments, and it creates a larger performance gap in the victim agent compared to competitive environments

## E. Continuous Action Space

As shown in (Wang et al., 2021a; Chen et al., 2022a), a series of continuous actions can be recognized by RL agents, which in turn suggests that our proposed supply-chain backdoor attack can be extended to continuous action space with minor adjustments:

- **Trigger action** The trigger action is defined in the same way as in discrete action space that a series of continuous actions $(a_{(1)}^t, \ldots, a_{(h)}^t)$.

- **Backdoor action** Since it is difficult and unnecessary to force the victim to precisely execute the backdoor action series with the exact float value in continuous action space, the backdoor action is defined as a set of action series $\{\hat{A}^b | \hat{A}^b := (\hat{a}_{(1)}^b, \ldots, \hat{a}_{(g)}^b), \|A^b - \hat{A}^b\|_\infty\}$ with tolerance to the target backdoor action $A^b = (a_{(1)}^b, \ldots, a_{(g)}^b)$.

To demonstrate the effectiveness of our proposed attack in continuous action space, we conduct experiments on Multi Particle Environments with the game Simple Push (Terry et al., 2021). This continuous environment involves two agents competing against each other, where the action is a five-dimensional vector with values between $[0, 1]$ indicating the amount of movement. We define the trigger action as four consecutive no-op actions $[0.0, 0.0, 0.0, 0.0, 0.0]$, the backdoor action as four no-op actions as well with tolerance $\|A^b - \hat{A}^b\|_\infty \leq 0.01$. The results utilizing PPO+CNN across various TIP are shown in Table 10, which illustrate the equal effectiveness of SCAB in continuous space.

## F. Stealthiness Evaluation

RL training typically diagnoses changes in behaviour by looking at bulk metrics (loss, reward, episode length, etc), as individually scrutinising transitions is labour-intensive and subject to significant variance due to noise. As shown in Table 11 and Figure 2, these results underscore our measure of stealthiness, as no noticeable differences are shown in the metrics.

*Table 8.* Results of Multi-player environment Volleyball Pong with PPO+CNN with $3\%$ TIP. The cases represent different settings of the attack. The results demonstrate that SCAB is effective in multi-agent environments.

| CASE | VICTIM | TRIGGER SUCCESS RATE | AVG. EPISODIC RETURN WITH TRIGGER PROPORTION | | | | |
|---|---|---|---|---|---|---|---|
| | | | 0% | 5% | 10% | 20% | 30% |
| 1 | SINGLE VICTIM | 42.1% | 12.5 | 10.3 | 9.3 | 8.6 | 6.7 |
| 2 | VICTIM 1 | 39.3% | 11.2 | 9.4 | 8.3 | 8.1 | 7.3 |
| | VICTIM 2 | 37.7% | 12.2 | 11.2 | 9.7 | 8.3 | 7.6 |

*Table 9.* Results of cooperative environment Volleyball Pong with PPO+CNN with various TIPs. The results demonstrate that SCAB is effective in cooperative environments.

| TIP | TRAINING-TIME | TESTING-TIME | | | | | |
|---|---|---|---|---|---|---|---|
| | AVG. EPISODIC RETURN | TRIGGER SUCCESS RATE | AVG. EPISODIC RETURN WITH TRIGGER PROPORTION | | | | |
| | | | 0% | 5% | 10% | 20% | 30% |
| 0% | 12.4 | N/A | 12.1 | 12.2 | 11.7 | 10.2 | 10.4 |
| 3% | 11.9 | 41.7% | 10.0 | 8.7 | 7.4 | 5.1 | 3.3 |
| 5% | 12.1 | 47.6% | 9.7 | 8.2 | 6.9 | 5.0 | 2.1 |

# G. Ablation Study: Trigger/Backdoor Action Pattern

The sequence of trigger and backdoor actions should be customizable, depending on the objective of the attack and the characteristics of the game. Some patterns are easier for an RL agent to recognize or learn. However, these factors should not be the sole consideration when choosing trigger or action patterns.

To explore the impact of trigger and backdoor action patterns on the performance of the supply-chain backdoor attack, we conduct experiments evaluating the Backdoor Success Rate against various patterns using Pong with PPO+LSTM. The effective actions in Pong are "move-up," "move-down," and "no-op," hence we exhaust all possible 27 action patterns with a length of 3 for the patterns. Selective results are presented in Table 12 and Table 13.

In summary, patterns with consecutive steps yield higher Trigger Success Rates for both trigger and backdoor actions. On the one hand, trigger action patterns with consecutive steps tend to produce more noticeable visual effects within a certain time frame, such as remaining still or executing relatively large movements. Consequently, such trigger actions are easier for RL agents to recognize, resulting in higher Trigger Success Rates. On the other hand, backdoor action patterns with consecutive steps are easier for RL agents to discern and replicate, thus yielding higher Trigger Success Rates.

# H. Ablation Study: Trigger Injection Time

To optimize the number of triggers injected during the training phase, we explore an alternative approach by framing the trigger injection process as an RL task. Instead of uniformly injecting triggers throughout the training phase, we train an RL agent to determine the optimal moments for trigger injection. The reward function for the training-time trigger network is designed as follows:

$$R^{\text{trg}}(s_i) = R^{\text{succ}}(s_i) + p\mathbb{1}(\pi^{\text{trg}}(s_i)) \; , \qquad (9)$$

which consists of the trigger success reward $R^{\text{succ}}(s_i)$ and the penalties for each timestep of trigger injection $p\mathbb{1}(\pi^{\text{trg}}(s_i))$.

Such an approach can achieve Trigger Success Rates generally exceeding $80\%$ while training. However, Trigger Success Rates drop significantly during testing without adversely affecting the victim's performance as shown in Table 14. This is attributed to the training-time trigger network effectively identifying moments when the victim is willing to execute the backdoor action without harming its own performance. This outcome contradicts the attack objective of compelling the victim to execute the backdoor action whenever the trigger occurs. Consequently, we consider the problem of constructing an effective and efficient strategy for injecting triggers during training time as an important future direction.

*Table 10.* Training- and Testing-time results for the game Simple Push with PPO+CNN model across varying TIP from 0% (no attack) to 5%.

| TIP | TRAINING-TIME | TESTING-TIME | | | | | |
| --- | --- | --- | --- | --- | --- | --- | --- |
| | AVG. EPISODIC RETURN | TRIGGER SUCCESS RATE | AVG. EPISODIC RETURN WITH TRIGGER PROPORTION | | | | |
| | | | 0% | 5% | 10% | 20% | 30% |
| 0% | -24.4 | N/A | -24.4 | -24.3 | -24.1 | -23.9 | -24.5 |
| 3% | -32.6 | 78.0% | -42.7 | -50.3 | -72.6 | -77.9 | -93.2 |
| 5% | -37.8 | 83.4% | -47.8 | -57.2 | -60.3 | -73.4 | -92.7 |

*Table 11.* Bulk metrics comparing clean and attacked training of CNN-PPO across various environments with a TIP of 1% with 95% confidence interval. Episodic Reward represents the Episodic *Cumulative* Reward.

| ENV. | POLICY LOSS | | EPISODIC LENGTH | | EPISODIC REWARD | |
| --- | --- | --- | --- | --- | --- | --- |
| | CLEAN | ATTACKED | CLEAN | ATTACKED | CLEAN | ATTACKED |
| PONG | $-0.037 \pm 0.004$ | $-0.029 \pm 0.006$ | $5177 \pm 231$ | $5241 \pm 264$ | $19.3 \pm 0.2$ | $19.5 \pm 0.1$ |
| BOXING | $-0.018 \pm 0.003$ | $-0.010 \pm 0.007$ | $541 \pm 67$ | $527 \pm 89$ | $91.4 \pm 0.8$ | $90.1 \pm 0.9$ |
| SURROUND | $-2.54 \times 10^{-4} \pm 1.3 \times 10^{-5}$ | $-1.23 \times 10^{-4} \pm 2.7 \times 10^{-5}$ | $1241 \pm 77$ | $1837 \pm 95$ | $9.3 \pm 0.2$ | $9.2 \pm 0.2$ |

## I. Fine-tuning Defense

We assess the effectiveness of the *fine-tuning* defense method against our proposed backdoor attacks by training the backdoored agent against normal players with additional steps aimed at *unlearning* the backdoor function in gameplay. As shown in Table 15, the fine-tuning defense demonstrates limited effectiveness. After $64,000$ steps, the Trigger Success Rate decreases by only $5\%$, while the victim's performance against backdoor-aware agents remains significantly compromised, dropping by $81.4\%$ under a $30\%$ trigger injection. This aligns with the assumption that fine-tuning is akin to reducing the TIP, while SCAB remains effective even with a low TIP as demonstrated.

Given that the victim's behavior remains consistent with that of a normal agent during both training and testing in the absence of the trigger as shown in experiments, detecting such an attack is also challenging.

*Table 12.* Trigger Success Rate against different backdoor action patterns in game Pong with PPO+LSTM and 3% TIP.

| TRIGGER ACTION PATTERN | BACKDOOR ACTION PATTERN | TRIGGER SUCCESS RATE |
|---|---|---|
| NOOP-NOOP-NOOP-NOOP | NOOP-NOOP-NOOP | 65.3% |
| NOOP-NOOP-NOOP-NOOP | NOOP-UP-DOWN | 22.2% |
| NOOP-NOOP-NOOP-NOOP | UP-UP-UP | 62.0% |
| NOOP-NOOP-NOOP-NOOP | DOWN-UP-DOWN | 24.1% |
| NOOP-NOOP-NOOP-NOOP | DOWN-DOWN-DOWN | 67.3% |

*Table 13.* Trigger Success Rate against different trigger action patterns in game Pong with PPO+LSTM and 3% TIP.

| TRIGGER ACTION PATTERN | BACKDOOR ACTION PATTERN | TRIGGER SUCCESS RATE |
|---|---|---|
| NOOP-NOOP-NOOP | DOWN-DOWN-DOWN-DOWN | 39.7% |
| DOWN-NOOP-UP | DOWN-DOWN-DOWN-DOWN | 14.5% |
| UP-UP-UP | DOWN-DOWN-DOWN-DOWN | 65.1% |
| UP-UP-DOWN | DOWN-DOWN-DOWN-DOWN | 14.5% |
| DOWN-DOWN-DOWN | DOWN-DOWN-DOWN-DOWN | 42.3% |

*Table 14.* Testing-time trigger success rates and average episodic returns (across a range of Trigger Proportions)

| ENVIRONMENT | LEARNING ALGORITHM | MODEL | TRIGGER SUCCESS RATE | AVG. EPISODIC RETURN WITH TRIGGER PROPORTION | | | | |
|---|---|---|---|---|---|---|---|---|
| | | | | 0% | 5% | 10% | 20% | 30% |
| PONG | PPO | CNN | 18.3% | 14.4 | 14.5 | 13.7 | 14.1 | 13.0 |
| | | LSTM | 26.2% | 13.2 | 12.2 | 12.7 | 11.8 | 9.4 |
| | DQN | CNN | 12.4% | 14.0 | 12.8 | 12.9 | 13.1 | 12.7 |
| | | LSTM | 17.6% | 14.1 | 14.1 | 12.7 | 13.4 | 12.8 |

*Table 15.* Impact of fine-tuning, where the victim is the same as the one described in Table 3 of Surround PPO LSTM.

| TRAINING-TIME | | TESTING-TIME | | | | | |
|---|---|---|---|---|---|---|---|
| ADDITIONAL FINE-TUNING STEPS | AVG. EPSODIC RETURN | TRIGGER SUCCESS RATE | NOT TRIGGERED | AVG. EPISODIC RETURN WITH TRIGGER PROPORTION | | | |
| | | | | 5% | 10% | 20% | 30% |
| 64000 | 8.9 | 78.5% | 9.5 | 7.0 | 4.9 | 2.7 | 1.3 |
| 76800 | 9.6 | 73.1% | 9.4 | 8.1 | 5.8 | 3.9 | 1.9 |
| 89600 | 9.5 | 72.3% | 9.6 | 8.4 | 5.7 | 3.1 | 2.4 |
| 102400 | 9.6 | 69.7% | 9.5 | 8.3 | 5.4 | 3.5 | 2.3 |
| 115200 | 9.7 | 71.7% | 9.5 | 8.5 | 6.0 | 4.2 | 3.0 |
| 128000 | 9.8 | 69.5% | 9.4 | 8.7 | 6.1 | 4.2 | 3.1 |

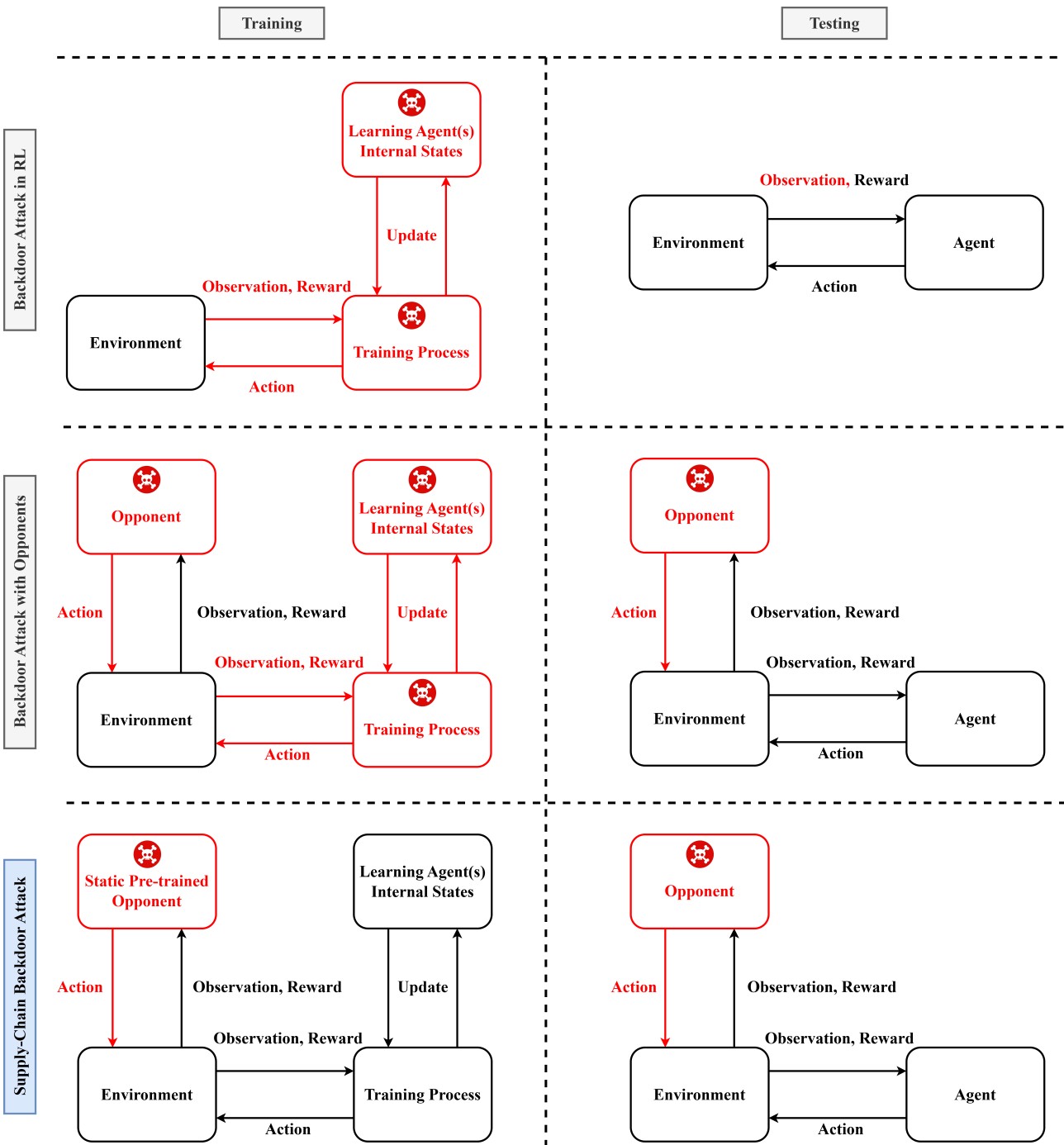

*Figure 3.* Comparative analysis of varied levels of access in terms of training and testing-time demands among backdoor attack techniques. The red line indicates write access to the corresponding parts or processes. The first row represents the backdoor attacks (Kiourti et al., 2020; Gong et al., 2024; Yu et al., 2022), the second row represents the backdoor attacks with an opponent agent (Wang et al., 2021a; Chen et al., 2022a; Wang et al., 2021b), and the third row represents our method.

