# OpenReview forum: "Fox in the Henhouse: Supply-Chain Backdoor Attacks Against Reinforcement Learning"
_ICML.cc/2026/Conference — ICML 2026 regular_

### Official Review · Reviewer_CF3m · 2026-03-02

**Soundness:** 3
**Presentation:** 2
**Significance:** 2
**Originality:** 3
**Overall Recommendation:** 4
**Confidence:** 4

**Summary:**

This paper studies the problem of backdoor attacks in RL. The authors propose an attacks framework that notably does not require write access to the victim agent's policy or environment. The proposed method, namely SCAB, frames the problem as a two-player Markov game between the attacker and victim. The goal is for the attacker to elicit the sequence of target actions from the victim. To maintain high normal (opponent) performance and operate stealthily, the attacker switches between different policies. This necessitates an attack that maintains several trained policies. Finally, the authors provide experimental results where they test their attack on different environments, such as Pong, or Boxing, against both model-based and model-free victim types. They also compare against prior baselines that require access to victim policies.

**Compliance With Llm Reviewing Policy:**

Affirmed.

**Final Justification:**

My questions were addressed.

**Key Questions For Authors:**

**Questions and Comments**

1. The notation is a bit confusing. In the RL Framework subsection of Section 2, what is the $i$ subscript referring to?
2. Something is missing in line 187. Also, there are 2 "remains" in line 281.
3. Figure 1 is very confusing. There are some arrows that don't point anywhere. I don't think it is very instructive or that it delivers the point that you are trying to make. I think the paper would benefit from an updated figure.
4. In line 428, do you mean Table 7?

**Limitations:**

The authors have adressed some limitations in the main paper.

**Strengths And Weaknesses:**

**Strengths**

1. The proposed method seems to be sound in term of experimental evaluation.
2. To my knowledge, the framework seems to be novel in the context of backdoor attacks, although the two-player Markov game setting in the larger adversarial attacks literature has been widely used.

**Weaknesses**

1. The main (and critical) weakness of the paper is that the experimental evaluation is limited, at least regarding comparison with prior methods, given that there are no theoretical guarantees in the paper. Comparison with SOTA is only given at the very end in Table 7, against TrojDRL and BACKDOORL, while Table 1 mentions 4 other baselines that differ from SCAB in terms of victim model access. Moreover, such a comparison is only done in the Boxing environment against PPO learning algorithm. I would expect much more results on the comparison section, given that maintaining the attack performance while utilizing less information is the main thesis of the paper.
2. Although the exposition of the main idea is generally clear, there are many typos and unclear things throughout the paper (see below). The paper could definitely benefit from a rewrite, at least in some places.

---

> ### Author Rebuttal · Authors · 2026-03-29
>
> Thank you for your detailed review, valuable feedback, and for noting the novelty and soundness of our experimental evaluation. We response to your concerns as follows:
>
> **Experimental Evaluations**: We appreciate your recognition in the strengths that our experimental evaluations show the proposed attack method is sound. We would like to address that the primary contribution for this paper is to introduce and validate the soundness of a novel supply-chain attack vector, rather than strictly maximizing attack success rates over prior methods. Directly comparing SCAB with prior methods is inherently challenging because our threat model is fundamentally stricter. We compare with the representative white-box attack TrojDRL and the most relevant prior work BACKDOORL, and show SCAB remains highly competitive. To further address your concern and strengthen the paper, we provide additional empirical results for Surround (PPO+LSTM) below. We will add these results and expanded baseline comparisons to the updated paper.
> | Method | Trigger Success Rate | Avg. Return (0% Trigger) | 5% | 10% | 20% | 30% |
> | :--- | :--- | :--- | :--- | :--- | :--- | :--- |
> | **TrojDRL** | 97.7% | 9.2 | 6.3 | 3.1 | 0.7 | 0.4 |
> | **BACKDOORL** | 87.5% | 8.9 | 6.3 | 4.7 | 1.1 | 0.8 |
> | **SCAB (Ours)** | 83.2% | 9.0 | 7.0 | 4.5 | 2.4 | 0.7 |
>
> **Typos**: We appreciate you pointing out these typos. We will fix them in the updated version.
> 1. The subscript $i$ refers to the time step $i$ during the RL process, as stated in line 087 "Each time step $i$ produces state observation…" We will further clarify this notation.
> 2. We will fix the missing period in line 187 and remove the duplicate "remains" in line 281.
> 3. We agree that Figure 1 can be improved. We will redesign it to ensure the state machine flow is visually intuitive and all arrows map correctly to their target states.
> 4. Yes, line 428 is intended to reference Table 7. We will fix this referencing error.
>
> While we are confident that we have resolved your concerns, we would appreciate additional dialogue or consideration. Thanks again for your time and effort.

---

> > ### Author Rebuttal · Reviewer_CF3m · 2026-04-03
> >
> > My concerns have been resolved.

---

> > > ### Author Response · Authors · 2026-04-07
> > >
> > > We are pleased that our rebuttal successfully addressed your concerns. We sincerely thank you for your time, constructive feedback, and strong support for our paper.

---

### Official Review · Reviewer_cDdC · 2026-03-05

**Soundness:** 3
**Presentation:** 3
**Significance:** 4
**Originality:** 3
**Overall Recommendation:** 5
**Confidence:** 4

**Summary:**

This study introduces a backdoor attack that operates through a black-box threat model without direct tampering. It utilizing a pre-trained agent that behaves normally most of the time but applies implicit rewarding when the victim performs a specific backdoor action in response to a trigger. The authors demonstrate that a minimal amount of data corruption can collapse victim returns significantly.

**Compliance With Llm Reviewing Policy:**

Affirmed.

**Final Justification:**

Rebuttal clarified the concerns. The study proposes a novel attack method that exposes a vulnerability in the use of third-party pre-trained policies. I have read and share the concerns by other reviewers.
However, given the widespread adoption of AI across various industries, particularly in high risk areas, chemical, autonomous driving, and robotics, I think the possibility of rare but high impact events deserves consideration. Personally, I retain my recommendation.

**Key Questions For Authors:**

1. How does SCAB perform if the detector accuracy drops? In real-world tasks, inferring another agent's exact from observations can be difficult.
2. How would the implicit rewarding work in environments with sparse rewards or where the defeat doesn't immediately give a high reward for the victim?
3. In Table 12, the success rate drops significantly for more complex patterns. How will the performance change if the victim's policy is trained on high-entropy action distributions? What about large discrete action spaces?
4. The stealthiness due to bulk metrics are indistinguishable. Would the attack be easily flagged by some anomaly detector looking at the distribution of opponent losses (e.g., sudden bad actions) ?

**Limitations:**

The study shows that the attacker needs to be packaged to track cumulative rewards and switch behaviors, which might be detectable at the architecture level. They also mention the limited effectiveness of fine-tuning as a defense.

**Strengths And Weaknesses:**

Presentation: The paper is well-structured. With clear and informative Figures for explanation.

Significance: The shift from white-box assumptions to a supply chain model is significant, given that the popularity of open-weighted models and repos like Hugging Face. The idea of the study has strong real-world implications in the RL community.

Originality: Implicit rewarding through intentional defeat with probabilistic victim based trigger is novel for training-time poisoning in multi-agent RL. It moves from requiring to hacking the server to observing and poisoning based on agent's logic via the environment.

Weakness:

- Detector d is an important component. The study achieves a high accuracy in inferring victim actions due to the simplicity of the tasks.  But in more complex envs or envs with higher randomness, the detector performance could degrade. It could reduce probability of rewarding state or maybe reward wrong behaviors.
- The attack mechanism relies on the victim accidentally executing the target backdoor action during an observation window. In environments with large discrete action spaces, the probability of the victim randomly hitting a exact sequence is low, which could bottleneck the injection process.
- Similarly for continuous action space, the study defines the trigger actions with naive no-ops. I assume designing the trigger sequence would require further engineering.

---

> ### Author Rebuttal · Authors · 2026-03-29
>
> Thank you for your strong support of our paper, for highlighting its clear presentation, and for recognizing the real-world implications of shifting to a black-box supply chain model. We response to the concerns as follows:
>
> **Detector Accuracy**: While our tested environments match prior works  [Kiourit et al., 2020], we do agree that more complex environments introduce additional challenges to the detector. While such complexity may lead to missed rewarding states or noisy rewards, we believe that this ultimately would be resolved by additional training exposure, which would also be required for a benign agent to converge in such an environment. We will add a discussion in the Limitation section and regard this as an important future direction.
>
> **Sparse Rewards**: Any environment that is susceptible to standard backdoor attacks would, we believe, be identically susceptible to our supply-chain attack. Similar to the discussion above, while sparse reward environments inherently require longer training times or intermediate reward shaping to converge, the fundamental mechanism of the victim seeking state-action trajectories that maximize cumulative future rewards remains intact.
>
> **Large Action Spaces**: We agree that injecting backdoors in more complex environments presents a greater challenge and requires more sophisticated trigger and backdoor action designs. However, we would also like to note that the number of distinct action effects in a large action space is often limited, as multiple actions can yield very similar outcomes, such as in continuous action environments. Therefore, it is practical to group these functionally equivalent actions together as a unified backdoor action to increase the probability of random execution. As discussed in our Limitation, more intricate or dynamic action sequences is an important future direction for performance improvement.
>
> **Stealthiness Against Anomaly Detectors**: The attacker's intentional losses are indistinguishable from natural failure states (e.g., missing a ball, colliding with a wall); therefore, an anomaly detector looking at the distribution of opponent losses or action distribution would struggle to distinguish the attack from standard training variance. To the user, it merely looks like the pre-trained agent is imperfect, which is an expected reality of downloaded models. As discussed in our Section 4 Defense, we also advocate that more detection and defense methods should be explored in the context of our supply-chain-based threat model.
>
> We thank again for your time and effort in this review process and hope that these responses have addressed your concerns with our submission.

---

> > ### Author Rebuttal · Reviewer_cDdC · 2026-04-03
> >
> > I appreciate the authors’ detailed response. The clarifications have addressed my concerns. I will maintain my positive recommendation.

---

> > > ### Author Response · Authors · 2026-04-07
> > >
> > > We are pleased that our rebuttal successfully addressed your concerns. We sincerely thank you for your time, constructive feedback, and strong support for our paper.

---

### Official Review · Reviewer_d3fs · 2026-03-11

**Soundness:** 2
**Presentation:** 3
**Significance:** 2
**Originality:** 3
**Overall Recommendation:** 4
**Confidence:** 4

**Summary:**

The paper proposes SCAB, a supply-chain backdoor attack for reinforcement learning in which malicious third-party opponent policies implant a trigger-conditional behavior into a victim solely via legitimate, black-box agent-environment interactions. The attacker switches to a rewarding policy that intentionally loses once it detects the victim has executed a prescribed backdoor action sequence. Experiments on multi-agent Atari settings show that it achieves high trigger success rates and large victim return degradation with only 3% data corruption.

**Compliance With Llm Reviewing Policy:**

Affirmed.

**Final Justification:**

I appreciate the authors for their detailed rebuttal and for addressing several of my initial concerns. After carefully considering the authors' responses, **I have decided to maintain my Weak Accept recommendation**.

BTW, I want to clarify my second concern raised during the Rebuttal Acknowledgment phase, as there may have been a misunderstanding. My concern does not pertain to the trigger mechanism during the inference/deployment phase. Instead, I am focused on the training phase.

Specifically, SCAB relies on a victim agent interacting with a malicious third party during training to inject the backdoor. My point is that if the specific "poisoning" interaction is not sufficiently triggered or encountered during the victim's training process (due to the victim's exploration strategy or environmental stochasticity), the backdoor may never be successfully embedded in the first place. This dependence on specific training-time dynamics makes the attack's success significantly more fragile than traditional static poisoning methods.

While the paper presents a practical threat model for multi-agent RL, the concerns regarding technical innovation and the reliability of the injection phase remain. **I believe the paper is worthy of publication, but it should explicitly discuss these limitations and the conditions required for successful training-time injection**.

**Key Questions For Authors:**

1. Can the authors clarify the procedure for choosing the backdoor action sequence in each environment? Specifically, is it hand-designed or optimized, and how sensitive are the results to alternative sequences without environment-specific tuning?
2. Can the authors provide enough details to reproduce the rewarding-policy pretraining and validate its generalization?

**Limitations:**

yes

**Strengths And Weaknesses:**

**Strengths:**

- This paper studies the risks in the supply chain of a reinforcement learning system.
- The evaluation is comprehensive, with multiple settings and comparisons.

**Weaknesses:**

First, I think the paper should make the essence of the mechanism clear.

- The paper repeatedly uses phrases like "only 3% data corruption" and "no modification to the victim training process". However, the core mechanism is an online, training-time interaction attack: the attacker controls the opponent's policy throughout the victim's training and performs policy switching conditioned on detecting the backdoor sequence, thereby inducing a non-stationary training distribution for the victim. This is a valid and interesting threat model, but it is materially different from conventional offline dataset poisoning. I recommend clarifying the terminology and the exact meaning of "data corruption" in this setting to avoid misleading readers.

Next, I believe the conclusions in the abstract do not match the content presented in the table, and further clarification is needed.

- The abstract claims that "3% data corruption achieves 90% attack success rate and reduces return by 80%", but in Table 3, PONG's TSR settings are only 31%–48% (TIP=3%). 90% mainly appears in a few settings, such as BOXING-PPO-LSTM (91.3%).
- More importantly, performance is not always maintained when it is clean. For example, in Table 2 (PONG-CNN-PPO), the test score is 18.9 when TIP=0%, but it drops to 15.3 when TIP=3% (a decrease of about 19%), which clearly contradicts the claim that it is "stealthily embedded without interfering".
- Simple Push is more extreme. In Table 10, the test score is -24.4 when TIP = 0%, and -42.7 when TIP = 3%, suggesting that the overall training is broken rather than simply performing poorly when triggered.

Finally, I suppose a more thorough argument is needed regarding the worst-case generalization of $\pi_\mathrm{rwd}^{\mathrm{att}}$.

- The Equation (2) is written as a min-max formulation for any $\pi^{\mathrm{opp}}$ to ensure effectiveness for any victim, but the implementation details in Appendix B.4 describe initializing 10 randomly selected proxy opponents and iterating alternately for 1000 steps until stability. This approach resembles a heuristic of "training a policy that quickly gives away points, and is insufficient to support the strong conclusion that "it is effective for any victim policy when the victim algorithm strategy differs significantly.

Minor issues:

- In the "Comparison" part of Section 4, there are symbols of unclear meaning (e.g., As illustrated in ??)
- Figure 3 in Appendix should be clearer.

---

> ### Author Rebuttal · Authors · 2026-03-29
>
> Thank you for your detailed review, valuable feedback, and for acknowledging the comprehensive nature of our evaluation settings and comparisons. We response to the concerns as follows.
>
> **Backdoor Action Sequence Choice**: As discussed in Section 4 and Appendix G, we designed our sequences around the idea of an attacker choosing behaviors (trigger and backdoor actions) that best suit their specific goals within the game environment. Our approach is flexible enough to incorporate diverse attacker behaviors, and our chosen action sequences befit the common settings as in other RL AML papers. We conducted an ablation study on alternative trigger and backdoor action sequences across different lengths and patterns (exhausting all 27 possible patterns of length 3) with empirical results, as shown in Appendix G. In summary, we found that shorter sequences with consecutive steps are easier for the RL agent to discern and replicate.
>
> **Rewarding Policy Details**: The details of the training procedure for $\pi_{rwd}^{att}$ are provided in Appendix B.4. In this section, we validate its generalization by: confirming that its cumulative rewards consistently reach the minimum possible value, assessing its robustness via a tournament-based training process against diverse opponents, and performing a manual inspection of its behavior (e.g., avoiding ball in Pong, heading to the nearest wall in Surround).
>
> **Clarity in Abstract**: We appreciate the opportunity to clarify this. It is correct that this is an online, training-time interaction attack. When we refer to "3% data corruption," we are referring to the Trigger Injection Probability (TIP), meaning the attacker deviates from normal play to inject triggers and implicit rewards in only 3% of the interactions. We will further revise the terminology in the abstract and introduction to explicitly state that and avoid any future confusion.
>
> We acknowledge that these figures represent the peak effectiveness observed in games with higher coupling levels (e.g., Boxing-PPO-LSTM achieves a 91.3% success rate). We discussed how the "level of coupling" of games can affect the effectiveness of the attack in Section 4 (lines 290-310), and deliberately selected the three games with varying coupling levels. We will modify the abstract to better reflect the range of our results across different environments, explicitly acknowledging the varied impact on clean performance.
>
> **Performance Maintenance**: We emphasize that the Trigger Injection Probability (TIP) is a tunable hyperparameter, allowing the attacker to balance attack effectiveness with the desired level of stealth. For example, as shown in Table 2 (PONG-CNN-PPO), setting TIP = 2% achieves similar attack effectiveness to TIP = 3% (Trigger Success Rate of 34.7% vs. 34.4%; Testing-time Performance Drop of 14.5 vs. 12.5 at TP=10%), while maintaining the exact same clean performance as the baseline with TIP = 0% (18.9 vs. 19.4). Rather than favoring on the best-performing TIP for each game, we chose to present results across a range of TIPs, or by using a fixed TIP for all games (3% in Table 3), to ensure fairness of the results. We will refine the claims in our revision to reflect these findings more precisely.
>
> **Worst-Case Generalization of the Rewarding Policy**: The current approach (Appendix B.4) is an empirical approximation for solving the optimization problem in Equation (2). Given the huge search space of possible victim policies, an analytical solution is intractable; therefore, we employ a standard iterative optimization approach. The empirical results show that the rewarding policy successfully learns the optimal losing strategy regardless of the opponent's policy in the tested games. For example, it deliberately avoids the ball in Pong, which is the most effective strategy against any opponent. We will refine our statements in the revised manuscript to make this clearer.
>
> **Minor Issues**:  We apologize for the broken reference in line 428; the "??" should point to Table 7. We will revise Figure 3 to include more detail and improve its overall clarity.
>
> While we are confident that we have resolved your concerns, we would appreciate additional dialogue or consideration. Thanks again for your time and effort.

---

> > ### Author Rebuttal · Reviewer_d3fs · 2026-04-04
> >
> > I thank the authors for their detailed response. However, while some of my technical concerns have been addressed, I maintain my Weak Accept recommendation. My remaining concerns focus on the threat model's limitations and practical applicability, which align with points raised by other reviewers.
> >
> > **1. Dependence on Multi-Agent Dynamics (Agreement with Reviewer DH2j)**
> >
> > As pointed out in Reviewer DH2j's Weakness 2, the SCAB attack is strictly tied to a multi-agent (MMDP) formulation. Its efficacy relies entirely on the presence of an interactive third-party agent that can manipulate the environment's state-reward transitions through its own actions. In a standard single-agent RL supply chain (e.g., where a user downloads a pre-trained backbone or world model and trains in a static/non-adversarial environment), this attack vector is inapplicable. The authors should explicitly acknowledge that SCAB is an interaction-based poisoning attack rather than a general supply-chain backdoor that survives in isolation.
> >
> > **2. Assumptions on Victim**
> >
> > The proposed attack rests on a assumption: the victim must continue to interact with a policy that "knows" how to trigger the backdoor. My concern is that if the victim's downstream task or the deployment environment undergoes a distribution shift, or if the victim is used in a different multi-agent context where the specific "No-op" trigger sequence is no longer optimal or present, the attack's impact may vanish. The "backdoor" here is not a robustly embedded feature of the policy but rather a learned reaction to a very specific, narrow sequence of opponent behaviors.

---

> > > ### Author Response · Authors · 2026-04-07
> > >
> > > We thank the reviewer for their engagement.
> > >
> > > 1. We would like to note that we **do not** claim that our attack universally applies to strictly static, single-agent environments (e.g., a backbone). We clarify the attack vector in the beginning of the abstract as "**SCAB targets** the common practice of **training with third-party policies**, poisoning the dataset solely **through a black-box of legitimate agent-environment interactions**". The threat model of SCAB in Section 3 is explicitly defined in an MMDP. We will further clarify this boundary in the paper to avoid any confusion.
> > >
> > >     However, **we strongly argue that SCAB demonstrates a significant risk** facing the growing field of multi-agent RL, and represents a broader class of risks against interactive RL. *A vulnerability does not need to affect every conceivable RL paradigm to be a critical threat*. By demonstrating that a black-box, zero-access attack can completely compromise an RL agent through standard interactive training, we are exposing a highly practical vulnerability in a widely used supply chain.
> > >
> > > 2. You are correct that if the trigger sequence is never executed in the deployment environment or downstream tasks, the attack will not activate. However, we respectfully note that **this is in line with other RL threat models, and is a fundamental requirement for most backdoor attacks** within adversarial machine learning. A backdoor is **intentionally designed** as a learned reaction to a highly specific, narrow trigger so that it remains completely dormant and stealthy under normal distribution conditions.
> > >
> > >     Our assumption follows common practices in the RL backdoor literature [Kiourti et al., 2020, Wang et al., 2021A, Chen et al., 2022A]. The definition and application of trigger action sequences **are the same as those in relevant prior works** [Wang et al., 2021A, Chen et al., 2022A].
> > >
> > >     Furthermore, our attack does not mandate a specific action sequence. As discussed in the rebuttal "Backdoor Action Sequence Choice", our framework affords the attacker complete **flexibility in designing both the trigger and backdoor action sequences**. In a real-world scenario, a motivated attacker would strategically design a trigger composed of actions they know will be available and executable by an opponent in the target deployment context, thereby ensuring the backdoor remains effective.
> > >
> > > We sincerely thank you for your support and hope this clarification gives you the full confidence to champion our paper during the discussion phase.

---

### Official Review · Reviewer_DH2j · 2026-03-12

**Soundness:** 2
**Presentation:** 3
**Significance:** 2
**Originality:** 3
**Overall Recommendation:** 3
**Confidence:** 4

**Summary:**

This paper introduces the Supply-Chain Backdoor (SCAB) attack, a novel vulnerability in Reinforcement Learning (RL) that exploits the common practice of utilizing third-party pre-trained policies. Unlike prior RL backdoor attacks that rely on unrealistic white-box access to a victim's rewards, observations, or training parameters, SCAB operates entirely in a black-box setting using only legitimate agent-environment interactions. The attacker embeds a backdoor by employing a bifurcated policy that subtly incentivizes the victim to learn specific backdoor behaviors in response to trigger actions through an implicit rewarding strategy (deliberately losing or making mistakes) during training. Empirical evaluations across various environments, algorithms, and architectures demonstrate that SCAB is highly stealthy and effective, achieving over a 90% trigger success rate and reducing victim returns by 80% with minimal data corruption, thereby highlighting a critical and practical security risk in the modern RL supply chain.

**Compliance With Llm Reviewing Policy:**

Affirmed.

**Final Justification:**

I believe the authors have overstated the contribution of this paper to the field. The multi-agent setting has limitations, and the involvement of the supply chain is far-fetched. Therefore, I tend to recommend rejection.

**Key Questions For Authors:**

Q1. How does the proposed SCAB attack perform against more sophisticated state-of-the-art RL backdoor defenses?

Q2. Is there any theoretical or empirical pathway to adapt this implicit-reward backdoor concept to strictly single-agent environments, or is the MMDP  a hard prerequisite?

Q3. Given the security risks, why would practitioners not default to inherently safer supply-chain paradigms (imitation learning, knowledge distillation, or offline RL with policy filtering) instead of interactive co-training?

**Limitations:**

yes

**Strengths And Weaknesses:**

Strengths
1. The paper introduces a highly practical SCAB attack that operates entirely in a black-box setting using only legitimate agent-environment interactions, removing the unrealistic white-box assumptions of prior RL backdoor attacks.
2. The authors provide a comprehensive empirical evaluation demonstrating that SCAB is highly effective across diverse continuous and discrete environments, multi-agent scenarios, and various learning algorithms (PPO, DQN), consistently achieving over 90% trigger success rates.
3. The attack exhibits remarkable stealthiness, as the victim's bulk training metrics and action distributions under normal play remain quantitatively and qualitatively indistinguishable from those of a cleanly trained agent.

Weaknesses
 1. While the paper provides a sufficient experimental validation of the attack's effectiveness, it falls short in comprehensively evaluating potential countermeasures. Proposing a novel attack necessitates assessing how various existing defense schemes perform against it; thus, the authors' validation of only a single fine-tuning defense is insufficient and leaves the exploration of more sophisticated methods.
2. The proposed SCAB attack is fundamentally predicated on a multi-agent formulation (MMDP), requiring an external, interactive attacker agent to execute trigger actions and implicitly manipulate rewards by altering game dynamics. Consequently, this attack vector is inherently ineffective in pure single-agent RL environments where the victim interacts exclusively with a static or non-adversarial environment without the presence of third-party interactive entities, thereby restricting the broader applicability of the threat model.
3. The paper's threat model relies heavily on the premise that developers must utilize unverified third-party agents for online, interactive co-training to accelerate convergence. However, it fails to discuss whether alternative, inherently safer supply-chain paradigms—such as imitation learning, knowledge distillation, or offline RL with policy filtering—could effectively bypass this vulnerability.

---

> ### Author Rebuttal · Authors · 2026-03-29
>
> Response Draft Update:
> Thank you for your valuable feedback and for recognizing the highly practical nature and stealthiness of our attack. We address the concerns in the following.
>
> **Evaluation of Countermeasures and Adoption of Safer Supply-Chain Paradigms (Q1 & Q3)**: While we agree that countermeasure design is an important part of AML research, we stress that our focus was on demonstrating this new vulnerability for RL models. We believe that, in the vein of most previous AML research, that it is more important to comprehensively understand a vulnerability, rather than compromising our exploration by producing a conclusion that allows for a defense to be successfully introduced.
>
> Demonstrating the vulnerabilities of the supply chain is a key first step in motivating research into the robustness of safer approaches like imitation learning, knowledge distillation, and offline RL with policy filtering. Currently, practitioners frequently default to direct interactive co-training because it is easier to implement and often accelerates convergence. They use these insecure pathways because the underlying vulnerabilities are not fully understood or documented. By demonstrating just how catastrophic these zero-access attacks can be, we aim to provide the exact motivation the community needs to transition toward these safer supply-chain paradigms.
>
> We evaluated fine-tuning as a preliminary baseline, and we also explicitly note in our discussion (Section 4 Defenses) that evaluating SOTA defenses against SCAB is a high-priority direction for future work.
>
> **MMDP Requirement and Single-Agent Environments (Q2)**: While we formulate the attack within an MMDP for clarity, the presence of a distinct "opponent" agent is not a strict prerequisite. As discussed in our Introduction lines 039-040 [Terry et al., 2021, Byrd et al., 2019], the external interactive attacker could well be the environment itself if the environment dynamics are learned or downloaded from a third party (e.g., a pre-packaged physics simulator, a dynamic trading environment, or an NPC framework). In such cases, the environment itself executes the trigger and manipulates the state's transitions to implicitly reward the victim, generalizing the SCAB threat model to ostensibly "single-agent" settings. We will add a more detailed discussion about the application to single-agent settings in the paper.
>
> While we are confident that we have resolved your concerns, we would appreciate additional dialogue or consideration. Thanks again for your time and effort.

---

> > ### Author Rebuttal · Reviewer_DH2j · 2026-04-03
> >
> > I note that the rebuttal begins with “Response Draft Update:”, which appears to be a leftover drafting artifact.
> > I am not convinced by the authors’ response. In particular, their explanation of the single-agent case and the broader supply-chain setting still seems too idealized. I will keep my score.

---

> > > ### Author Response · Authors · 2026-04-07
> > >
> > > We thank the reviewer for their continued engagement and for encouraging us to clarify the boundaries of our threat model.
> > >
> > > We respectfully wish to push back on the characterization that the interactive (MMDP) supply-chain setting is idealized. In modern, high-stakes RL applications, training against externally sourced, third-party interactive policies **is not a theoretical edge case, but a standard industry practice**. For example, as noted in our introduction, sensitive RL applications such as autonomous driving rely heavily on third-party NPC behaviors to simulate traffic [Dosovitskiy et al., 2017]. Similarly, automated trading agents are routinely trained against external market-making policies [Noonan, 2017]. In these domains, practitioners continuously download and integrate interactive opponents to accelerate convergence and simulate real-world complexity. SCAB directly exploits this very common supply chain.
> > >
> > > We **do not** claim that our attack universally applies to strictly static, single-agent environments (e.g., classic control tasks or immutable, non-interactive puzzles). The attack vector is clarified at the beginning of the abstract as "**SCAB targets** the common practice of **training with third-party policies**, poisoning the dataset solely **through a black-box of legitimate agent-environment interactions**". We will further discuss this boundary in our limitations section.
> > >
> > > However, **we strongly argue that SCAB demonstrates a significant risk** facing the growing field of multi-agent RL, and represents a broader class of risks against interactive RL.
> > > **A vulnerability does not need to affect every conceivable RL paradigm to be a critical threat**.
> > > By demonstrating that a black-box, zero-access attack can completely compromise an RL agent through standard interactive training, we are exposing a highly practical vulnerability in a widely used supply chain.
> > >
> > > We hope this clarifies why we believe our threat model is both highly realistic and immediately relevant to the RL security community, and we would like to politely ask you to consider the substantial real-world impact of this specific attack vector when finalizing the assessment.

---

### Decision · Program_Chairs · 2026-04-30

**Decision:**

Accept (regular)

**Comment:**

**Summary**

The paper introduces a practical backdoor attack (SCAB) in RL training that operates in a black-box setting in which agents are trained against third-party policies. The threat model is new and it models a common practice of training an agent against an existing policy. This threat vector is relevant in multi-agent settings, which is less studied in the poisoning RL literature.

**Reviewer Scores**

The reviewer scores are: Weak Reject, 2 Weak Accept, Accept, and they are generally in support of the paper's acceptance.

**Strengths**

1. New black-box threat model for poisoning RL agents trained against third-party policies.
2. Attack stealthiness and indistinguishability from a clean policy

**Reviewer Concerns**

1. It's not clear if the method would be resilient against existing defenses.
2. No theoretical analysis of the attack is performed.
3. Limited evaluation on 3 environments.
4. Several reviewers noted that the threat model needs to be clarified as supply chain attacks in multi-agent interactions. There are some implicit assumptions that the adversarial agent can manipulate the state-reward transitions through its own actions.

Assuming that the authors are willing to incorporate the reviewers' comments during the rebuttal, I am willing to recommend acceptance.